# Mesospheric winds measured by MF radar with Full Correlation Analysis: error properties and impacts on studies of wind variance

Maude Gibbins[1, 2] and Andrew J. Kavanagh[2,3]

[1]Formerly the University of Cambridge, Cambridge, UK
[2]British Antarctic Survey, High Cross, Madingley Road, Cambridge, CB3 0ET
[3]Visiting Scientist at RAL Space, Rutherford Appleton Laboratory, Harwell, Oxford, OX110QX

*Correspondence to*: Andrew J. Kavanagh (andkav@bas.ac.uk)

**Abstract.** The mesosphere is one of the most difficult parts of the atmosphere to sample; too high for balloon measurements and too low for in-situ satellites. Consequently there is a reliance on remote sensing (either from the ground or from space) to diagnose this region. Ground based radars have been used since the second half of the 20th century to probe the dynamics of the mesosphere; Medium Frequency (MF) radars provide estimates of the horizontal wind fields and are still used to analyse tidal structures and planetary waves that modulate the meridional and zonal winds. The variance of the winds has traditionally been linked qualitatively to the occurrence of gravity waves. In this paper the method of wind retrieval (full correlation analysis) employed by MF radars is considered with reference to two systems in Antarctica at different latitude (Halley at 76°S and Rothera at 67°S). It is shown that the width of the velocity distribution and occurrence of 'outliers' is related to the measured levels of anisotropy in the received signal pattern. The magnitude of the error distribution, as represented by the wind variance, varies with both insolation levels and geomagnetic activity. Thus it is demonstrated that for these two radars the influence of gravity waves may not be the primary mechanism that controls the overall variance.

## 1. Introduction

Located around 50 to 100 km altitude above the Earth's surface, the mesosphere is one of the most difficult places to directly study; too low for satellites to pass through and too high for meteorological balloons. Apart from sporadic rocket experiments, most information on the dynamics and chemistry of this region has been via remote sensing: either satellites at much higher altitudes or ground based instruments such as radars. One type of radar that has been used extensively to probe the mesosphere is the medium frequency (MF) radar. Originally developed to study changes in electron density in the lower ionosphere (e.g. Gardner and Pawsey, 1953), the MF radar receives signals that are partially reflected from gradients in the weak D-region plasma that co-exists with the neutral atmosphere.

At sufficiently low altitudes (below ~95 km) the electron density is usually small enough that effects of the refractive index on the signal speed are negligible. In this height range, the motion of the plasma is dominated by the background neutral wind

such that careful analysis of the returned signal measured on spaced receivers can provide a means of estimating that velocity. Above 95 km as the plasma density increases the medium frequency waves (a few MHz) are refracted such that the height of partial reflection cannot be accurately assessed, this is particularly the case when there is enhanced geomagnetic activity and the radar is close to the auroral zone. With increasing altitude the plasma motion is no longer dominated by the neutral wind, rather the ionospheric electric field starts to become more important, particularly during large geomagnetic storms; this also

limits the useful height range of the radar (Reid, 1983)

The dynamics of the mesosphere are dominated by a number of strong wave modes. Thermally driven solar tides propagate from lower altitudes, their amplitudes often maximising in the mesosphere. Tides occur with periods at harmonics of the solar day (e.g. 24 hour, 12 hour, 8 hour) and will dominate the wind field on this timescale (e.g. Manson et al., 1989; Forbes, 1990) Planetary waves generated in the troposphere also penetrate into the mesosphere modulating the winds and temperatures over

periods of days to weeks. At much smaller scales gravity waves play an important role in the dynamics of the neutral atmosphere (e.g. Fritts, 1984); they are generated by wind over orography, by convective storms and by wind shears (such as the edge of jets). These buoyancy waves carry energy and momentum through the atmosphere and when they break they deposit that momentum into the mean flow acting either as a break or to accelerate the flow. This mean flow is part of a large scale circulation pattern that links the two poles: rising in the summer hemisphere and down welling in the winter in the polar

vortex. Hence these waves play an important role in the atmosphere; however, due to their size they tend to be unresolved by general circulation models and so their effects are parametrised in the models. Getting this parameterization right is important for our knowledge of the dynamics and chemistry of the atmosphere and consequently it is essential to understand the properties and propagation of these waves through the atmosphere.

Just as for the models the scale sizes of gravity waves renders them invisible to MF radars, which are sampling and averaging

wind measurements over a large portion of the sky. Although the radars cannot resolve them directly it is assumed that the waves influence the estimated horizontal winds by increasing the variance of the measured winds, essentially introducing fluctuations about the mean observed wind. Consequently the variance of the horizontal wind velocity determined by MF radars is taken as a proxy for gravity wave activity. This has enabled researchers to build up climatological patterns for the occurrence of gravity waves in the mesosphere (e.g. Hibbins et al., 2007).

The results of the study presented here call into question the validity of assuming that gravity waves are the principle cause of variance in MF radar measurements. It is found that the dominant cause of high variance is linked to the solar illumination of the mesosphere and consequently changes in plasma density. The influence of gravity waves is not ruled out but their role in the variance is shown to be somewhat smaller than past work might have shown.

## 2. Instrumentation

The British Antarctic Survey operates two medium frequency (MF) radars at their Antarctic research stations, one at Rothera (67°S, 68°W) on the Antarctic peninsula and the other at Halley (76°S, 26°W) on the Brunt ice shelf. Figure 1 shows the locations of the two stations (red squares). The dashed black circles represent estimates of the statistical location of the edge of the polar vortex through winter from May (inner circle) to August (outer circle) (taken from Zhang et al., 2017). The shaded green region represents the extent of the quiet-time auroral oval (determined from Holzworth and Meng, 1975), where we have assumed local midnight lies between Rothera and Halley for the purposes of illustration.

Both systems are coherent, spaced-antenna wind profilers that measure the horizontal neutral winds in the mesosphere and lower thermosphere using the full-correlation analysis technique (Briggs, 1984) where the transmitted signal undergoes partial reflection from density gradients in the weakly ionized atmosphere. The Rothera radar is a joint project with GATS Inc. whereas the Halley radar is owned by BAS. It should be noted that at the time of writing the radar at Halley is non-operational due to the seasonal closure of Halley station in response to the instability of the Brunt ice shelf.

The difference in location of these radars has two main implications relevant to this study: Rothera is located at a region of intense gravity wave generation in the lower atmosphere, where winds passing over the Antarctic peninsula (and the Andes to the north) will generate mountain waves. Large convective storms that pass through the Drake Passage also give rise to gravity waves. During the winter the site is close to the edge of the polar vortex, which can also produce gravity waves (e.g. Beldon and Mitchell). Conversely, Halley is in a quieter region and well within the polar vortex and as such we would not expect to see as much gravity wave activity there (e.g. Espy et al., 2006). Halley is on the edge of the auroral zone, so a geomagnetic influence on the mesosphere may be more significant than at Rothera.

The two MF radars have some intrinsic differences: the radar at Halley (2.7 MHz) operates at a higher frequency than Rothera (1.98 MHz), and is much more powerful (~120 kW vs. 25 kW), resulting in increased reflected signal and more viable data in the lower range gates where there plasma gradients are weaker. Table 1 summarises some of the basic properties of the two radars.

In both data sets the wind measurements have a vertical resolution of 4km, oversampled in 2 km range gates giving a total of 24 altitude steps. Due to the increased ionisation in the upper mesosphere, data coverage is much better in the higher range gates than in the lower, where coverage is patchy and mostly only available in the daytime and during the summer months. Due to the nature of the technique and the location of measurement, there is much variability in the data coverage, with data gaps ranging from single missing data points to several months.

This study uses the horizontal wind velocity data obtained from the two radars. Winds are derived from the radar signal using the full correlation analysis technique (FCA) outlined by Briggs (1984). This technique uses spaced receiver antennas to estimate the bulk motion of a time-evolving pattern of reflected wave scatter from the atmosphere. For both Halley and Rothera radars the three receiver antennas are aligned north-south and east west in a 'L' configuration (though off-orthogonal arrangements are used at other radar sites). There are several factors linked to the received signal that limit the data analysis.

No analysis is performed for: a signal to noise level below -8 dB; a cross-correlation function magnitude of 0.2 and a normalized time discrepancy of less than 35%.

## 3.  Data properties

Figure 2 shows example time series of the wind data from (a) Halley and (b) Rothera for three altitude gates (98 km, 82 km and 74 km). Due to a timing issue at Rothera the range gates are offset by ~4.5 km. Blue (red) dots represent the zonal (meridional) winds. These illustrate some of the inherent properties of the data. Most measurements lie between -100 and 100 ms$^{-1}$ for this interval (87% of the data at Rothera, 76% at Halley) with seemingly random outliers. An oscillating signal is present in panels (b), (d) and (e); this is the semi-diurnal tide, which maximises in the high mesosphere. Tides are a major component of mesospheric winds in both the zonal and meridional directions. The dominance of tidal modes depends on location and time of year: for Halley the magnitudes of the diurnal and semidiurnal tides maximise in the summer months (Hibbins et al., 2006); for Rothera, the diurnal tide maximises in summer whilst the semi-diurnal peaks in winter (Hibbins et al., 2007). The tides tend to have wind amplitudes of a few tens of m/s, which can be additive depending on the phases of the tides. This still leaves a considerable amount of data that would be described as 'outliers'.

### 3.1 Outliers and error distribution

Large velocity outliers are common in both the meridional and zonal winds and can reach magnitudes greater than 100 ms$^{-1}$, several orders of magnitude above what would be considered a normal range of wind speeds. The presence of such outliers is not limited to the radars discussed here:  publications using MF radar data often mention an outlier-removal step such as median filtering (Dowdy et al., 2001), removal based on a running mean (Dowdy et al, 2007), and simple exclusion of wind speeds greater than a given threshold (Holdsworth and Reid, 2004). In all of these studies, the nature of the excluded data is not discussed, and justification for the outlier removal step, where given, is to remove data that is of poor quality due to a low signal-to-noise ratio (e.g Thayaparan et al., 1995).

Figure 3 shows the distribution of all wind speeds (zonal and meridional) measured by each of the two radars for the entire dataset. Panel (a) shows the probability distribution function (PDF) of a given velocity for data from Rothera (blue) and Halley (red). All velocity values are included: both zonal and meridional winds from all range gates. The PDF for each radar has a double hump, centred on the zero velocity, caused by the tidal nature of the wind; the rate of change of the wind will be lower where the tides have extrema, with a smaller number of data points appearing around zero velocity. The peaks represent some form of average magnitude of all significant tidal modes in the data set.  Panel (b) focuses on the right hand tail of the distribution, presenting the data on a logarithmic scale. The tails of the distribution from both radars are virtually identical, above ~200 ms$^{-1}$ the data are well represented by a Lorentz (or Cauchy) distribution. This distribution is defined by

$$P(X = x) = \frac{b}{\pi(x^2+b^2)}$$

(1)

where b is a parameter describing the width of the distribution. A value of b=5.7 was found to match the observed distribution of high wind speeds for both radars.

## 3.2 Relationship with pattern axial ratio

The full-correlation analysis (FCA) method for calculating wind speeds is based on the correlation of patterns in the radar reflection from the ionised portion of the mesosphere that decay both in space and time. Wind speeds are calculated by performing lagged correlations between the signals received by the spaced antennae and using the lag times to derive the velocity of the overall motion of the pattern. The spatial components of the surfaces of constant correlation are described by an ellipse, i.e.

$$\rho(\xi, \eta, \tau = 0) = A\xi^2 + B\eta^2 + 2H\xi\eta = \text{constant}$$

(2)

where A, B, and H are coefficients of the ellipse, $\tau$ is the lag time, and $\xi$ and $\eta$ are distances in the x- and y-directions. As well as the derived wind speed, the radar data used in this study also contains properties describing the ellipse defined by equation 2. A fully detailed description of FCA can be found in Briggs (1984), but we have included a basic description in the appendix.

Figure 4 shows a relationship between increasing wind velocity and high axial ratio of the FCA ellipse for measurements from the Rothera radar. The contours show the rapidly declining count density (counts ratio$^{-1}$m$^{-1}$s) at higher values; the data are sorted into equally spaced logarithmic bins. Contour lines are presented at order of magnitude changes. For each velocity direction ((a) west, (b) east, (c) south and (d) north), there is a relationship between increasing velocity and axial relationship: high velocity is associated with high axial ratio. The data in fig. 4 are from all altitude ranges from the Rothera radar; limiting the altitude range produces the same results indicating that the relationship between velocity and axial ratio appears to be independent of altitude for the range of heights that the radars measure. The pattern remains the same when data from Halley are used.

Yamazaki et al. (2000) discussed the relationship between high axial ratio and extremely high wind measurements. They performed a study of the FCA technique that involved storing and manipulating raw radar signals to observe the behaviour on the outcomes of the FCA calculation. Yamazaki et al. (2000) found that by artificially clipping raw radar signals before applying FCA, the resulting wind speeds sometimes reach extremely high values at low levels of saturation. They point out that the source is the final step in the FCA calculation, which involves solving the following system of linear equations:

$$Av_x + Hv_y = -F$$

(3)

$$Hv_x + Bv_y = -G$$

(4)

where A, B, H, F and G are derived from the correlation lag times. The solution to equations (3) and (4) is given by:

$$\begin{pmatrix} v_x \\ v_y \end{pmatrix} = \frac{1}{AB-H^2} \begin{pmatrix} B & -H \\ -H & A \end{pmatrix} \begin{pmatrix} -F \\ -G \end{pmatrix} \qquad (5)$$

which contains a division by the determinant of the coefficient matrix, $AB - H^2 \equiv \Delta$. Yamazaki et al. (2000) found that the extremely high values occurred when $AB \approx H^2$, meaning $\Delta \approx 0$. $AB \approx H^2$ implies that equation 2 describes an ellipse with a high axial ratio, i.e. the pattern has a high degree of anisotropy. Where $AB < H^2$ equation 2 describes a hyperbola rather than an ellipse. A hyperbolic surface of constant correlation is not physical in this context: this suggests that, in a particular direction, the correlation of the signal increases with increasing separation. Indeed, hyperbolic contours is one of the rejection criteria listed by Briggs (1984) (albeit with a misstatement of the direction of the inequality). Interestingly, Yamazaki et al. (2000) present scenarios with both $AB > H^2$ and $AB < H^2$ for their unsaturated results, suggesting that this rejection criteria was not included in their analysis.

Very high wind values caused by $\Delta \approx 0$ suggest an error distribution due to division of two Gaussian distributed variables. This results in a Lorentz distribution, which is consistent with the error distribution observed in the measured wind velocities.

Further evidence of a relationship between errors in the wind measurements and a high axial ratio is found by observing the 2D distribution of wind speeds measured when the axial ratio is high (>5). Figure 5 shows this distribution for both the Halley and Rothera radars, with data considered across all altitudes. There is a distinct geometric pattern, which is similar between both radars and seems to be an artefact of the 3-receiver antenna arrangement. This implies that the wind data involve errors related to the measurement technique. This also shows that simply filtering out high wind speeds from the data will not eliminate these errors.

### 3.2 Impact on measured wind variability

Studies of the dynamics of the mesosphere and lower thermosphere often focus on gravity waves, due to their importance in carrying energy and momentum through the system (e.g. Brasseur and Solomon, 2005). Although MF radars cannot resolve gravity waves, the variance of the calculated winds are taken to be a qualitative measure of the occurrence of gravity waves. In this section the impact on such studies of the error properties presented in the previous section is considered.

In the literature, several analysis procedures are used to study high frequency variations in MF radar wind data. Many studies start by taking means of the data over certain time intervals (ranging from 10 minutes to 1 hour), then perform Fourier or wavelet analysis to separate the variance into different period bands (Dowdy et al., 2001; Dowd et al., 2007; Hoffman et al., 2011; Hoffman et al., 2010; Isler and Fritts, 1996; Meek et al., 1985; Nakamura et al., 1993; Vincent and Fritts, 1987). Other studies simply take the variance of the raw data, after fitting and removing components due to tides and lower-frequency mean winds (Hibbins et al., 2007, Thayaparan et al., 1995).

In all of these cases, random errors of individual measurements could have some impact on the observed variances. However, this relationship is most easy to interpret in the case of binned raw data variances, which directly measures all signal and noise

with time frequencies under the time period of the binning. Therefore raw hourly variances are considered in this investigation, with outliers removed by applying a conservative acceptance threshold of 150 ms$^{-1}$ (a necessary step to avoid extreme outliers having a disproportionate influence). This choice represents the minimum possible data manipulation, avoiding steps such as fitting and removing tidal signals or interpolating to even time steps for Fourier or wavelet decomposition, all of which can potentially introduce biases into the data (Mudelsee, 2010). In addition, an hour represents the minimum time period over which a sufficient number of data points is usually present, while the influence of signals at tidal frequencies and below remains negligible.

By assuming that the error is Lorentz-distributed and that all values of velocity with magnitude greater than some limit, T, are always due to this error, the b parameter of the distribution, and the expected impact of the errors on the hourly variance parameter can be estimated. Integrating equation 1 between T and –T and solving in terms of b gives:

$$b = T \, \tan(\frac{\pi}{2} P_T) \tag{6}$$

where $P_T$ is the proportion of the data that falls outside of the threshold T and is calculated numerically from the data. We set T = 300 ms$^{-1}$ which represents a conservative choice, well above 'normal' wind speeds of under 200 ms-1 (estimated from panel (b) of fig. 4), so we can be confident that values in this range are not due to true wind speeds. To estimate the expected hourly variance due to this error distribution we generate data from

$$x_i = b \frac{R_1}{R_2} + \sigma R_3 \tag{7}$$

where R1, R2, and R3 are standard normally distributed variables and σ is set at 30 ms-1. The first term of this expression gives a Lorentz distribution with parameter b, and the second term allows for variance due to changes in the actual wind speed, for example due to tides. For each altitude simulated data was generated from equation 7 (where i = 1 million and is the length of the simulated time series) and the mean variance for velocities below 150 ms-1 measured using a monte-carlo method with 100 iterations.

This allows direct comparison between observed variance and expected variance based on the fitted Lorentz error distribution. The hourly mean of the axial ratio is also considered as an additional proxy for the expected accuracy of the data, since we have seen that a higher axial corresponds to a larger error distribution. The observed variance and axial ratio are averages of the hourly means that were calculated from data with wind speed < 150 m/s.

Figure 6 shows that the vertical profiles of hourly mean zonal variance (black), Lorentz-predicted variance based on the number of outliers (black circles) determined from the process outlined above, and mean axial ratio (red) for both Rothera (panel a) and Halley (panel b).

The lines diverge most at the highest altitudes suggesting that perhaps true wind variance contributes more at these altitudes. This might be expected since as gravity waves propagate upwards, they grow in amplitude as the local density decreases and

then they break, depositing their momentum into the background wind flow (e.g. Kelley, 2009). The actual height of breaking will depend strongly on the spectrum of waves that are present. The tides may also play a role, since the tidal amplitudes maximise at the upper end of the radar range such that the position of the peak velocity could skew to higher values not captured in the simulation.

However caution must always be exercised when considering the wind values above ~95 km as there are three ways in which the winds can be modified by geomagnetic effects:

- an overly enhanced D-layer will increase the local refractive index such that the radar beam slows down and refracts such that one can no longer be sure of the height of returned echoes.
- at higher altitudes >105 km the local electric field can start to pay a role and the electron density structures from which
the radar beam scatters will no longer drift with the local wind background.
- A second factor associated with increased electron density is attenuation of the beam; An increase of electron density coupled with the high electron-neutral collision frequency in the mesosphere results in loss of radar signal (e.g. Kavanagh et al., 2018)

Below we consider other factors that might contribute to the variance of the wind speed that are not linked to intrinsic wind
properties or wave features.

## 4. Causes of high measured wind variance

In this section we consider two factors that are likely to contribute to the variance of the wind data and which would play a role in the varying height distribution. The underlying cause for both factor is changing signal qualities driven by changes in the scattering efficiency in the ionized mesosphere: ion densities also change dramatically with height, which affects the
230 scattering quality, in turn affecting the magnitude of the error distribution. Published studies of gravity wave climatologies assume either explicitly or implicitly that varying data quality is not the cause of the observed vertical profiles. No justification for this assumption was found in any of the studies referenced in this report, beyond "inspection of the data" (Dowdy et al., 2001), and the observation that availability of data does not change (Thayaparan et al., 1995).

### 4.1 Solar Illumination

If one adopts the interpretation that measured variance is dominated by changes in the scatter quality, many of the observed daily and seasonal trends in variance may be readily understood. Since the dominant source of ionisation in the mesosphere is photo-ionisation due to sunlight (and ionisation levels decay considerably during the night), levels of solar illumination can be expected to have a strong impact on the data quality.

Figure 7 (a) shows the solar elevation angle as a function of local time, for the average day in four months spaced throughout
the year to provide maximum contrast, calculated for Rothera. Panel (b) shows the zonal wind variance for the average day as
a function of local time across the altitude range of 88.5 to 90.5; (c) shows the same for the meridional wind, Both show a
dependence on solar elevation angle; as the elevation angle increases the average variance decreases. The background level of
variance tracks with season: summer months have lower values of variance than the winter months. This is interpreted as a
response to the changing levels of ionisation. The months with lowest solar elevation have a significant asymmetry, with
variance being slow to recover at dusk; this fits with differences in detachment and recombination rates of atmospheric
chemical constituents in the low ionosphere (e.g. Collis, and Rietveld, 1990) where ionized molecules persist after the source
of illumination is removed.  The lower meridional wind variance reflects the fact that meridional winds tend to be weaker than
their zonal counterparts.

To examine the relationship between solar elevation angle and variance further, the mean variance for each hour in the day
and each month was taken, during which time solar elevation angle is approximately constant. This revealed an inverse
relationship between solar elevation angle and variance that persists across all altitudes at both radar sites. The magnitude of
the relationship is lowest at middle altitudes (70 - 85 km), increasing above and below this range.

Figure 8 shows examples of this relationship at three altitudes (92.5 km,80.5 km and 72.5 km) at Rothera (blue) and Halley
(orange) for both the zonal (a-c) and meridional (d-f) winds. Each point represents one of the 24x12 hour-month combinations.
There is a clear change in the variance that occurs with solar elevation angle; between ~-10 and 10 degrees there is a relatively
sharp transition that separates high variance values at negative solar elevation (sun below the horizon), from low variances at
positive angles (sun above the horizon). The transition begins at a smaller elevation angle (~-9 degrees) at the higher altitude
(panels a and d) than in the two lower altitudes (~-15 degrees) (panels b and c). Given that the solar elevation angle is calculated
for the surface of the Earth, this effect may be due to the shadow height of the Earth.

Another effect is the distribution of variance values during darkness compared with the sunlit data. Lower altitudes (panels c
and f) have a much wider spread of variance with negative solar zenith angle. There is a difference between the radars at the
highest altitude for the zonal wind (a) but not for the meridional wind (d). The cause for this is not known butcould be a result
of a fundamental difference in the stability of the ionosphere in darkness at the higher latitude since solar illumination is not
the only source of ionization.

In order to confirm that the observed variance changes with sunlight are indeed a function of differing error distributions, the
relationship between axial ratio and number of outliers (a), and both zonal (b) and meridional (c) variance is shown in Figure
9. In these plots, every point represents a separate hour-month- altitude combination, separating out the differing responses to
solar elevation angle.

A strong correlation between axial ratio and both number of outliers and hourly variance is observed, providing evidence that
the changes in variance result at least in large part from changing levels of data quality, rather than real wind features. At this
stage we note that the two radars display different relationships, the reason for this is not clear though the Halley radar does

operate at a different frequency and is much higher power than the Rothera radar which might affect the data selection prior to calculating the winds via the full correlation analysis. For both radars, the shapes of the relationships between parameters are consistent such that the relationship between variance and number of outliers is linear.

Given the relationship between solar elevation angle and variance presented in the previous section, it follows that an annual trend in variance would be seen due to the seasonal changes in sunlight levels. This is characterized by an increase in variance during the winter and a decrease during the summer.

Figure 10 shows the change in zonal (b) and meridional (c) variance throughout the year at Rothera, along with the mean solar elevation angle above the horizon (a). Indeed, a strong annual cycle corresponding to sunlight levels is seen, with the highest

variances occurring at the lowest and highest altitudes during winter. This fits with the lower and upper bounds being regions of less and more data. It is worth noting that the pattern is not uniform, even with smoothing (a 15-day running mean) applied. This does suggest that other factors contribute to the climatology and leaves room for natural wind turbulence to play a role once these have been accounted for.

## 4.2 Geomagnetic Activity

In general the plasma density in the polar mesosphere increases when geomagnetic activity is high due to increased charged particle precipitation. A change in the plasma density will affect the strength of the reflected radar signal (e.g. Kavanagh et al., 2018), which in turn could alter the measured wind variance through changing the amount of data in a given period. Halley in particular is located on the edge of the auroral zone, where it is known that geomagnetic activity affects the chemistry of the mesosphere (Brasseur and Solomon, 2005).

To probe this relationship the Auroral Electroject (AE) index is used as a measure of geomagnetic activity. This index is derived from geomagnetic variations in the horizontal component of the magnetic field observed by 10 to 13 stations in the auroral zone in the northern hemisphere. The AE index is the difference between the largest and smallest values detected by these stations, produced at 1-minute resolution. It responds most strongly to the substorm cycle, where energy is loaded in the magnetotail from the solar wind, and then released earthward generating the auroral electroject and auroral displays. Although

there can be quite drastic differences in the local scale structure, magnitude and positioning of auroral forms (and the underlying magnetic topology), between the poles, in a statistical sense the AE index will still be representative of geomagnetic activity in the south.

Figure 11, panel (a), shows the cross correlation between the daily averaged AE index and the daily averaged zonal wind variance measured at Halley at three altitudes: 90 km, 80 km and 70 km. Each of the data sets have been normalized such that

their autocorrelations equal one at the zero lag and lie between 1and -1. At each altitude there is an annual cycle in the correlation, though the value of the coefficient is relatively small (<0.2). This cycle is due to the seasonal variations of both the variance and the AE index; the variability of the AE index is driven by changes in solar wind activity, but the coupling to Earth's magnetic environment has a seasonal component known as the Russell-McPherron effect (Russell and McPherron,

1973), whereby the coupling maximises around the equinoxes. Figure 10 illustrated that there is a seasonal pattern in the variance, which matches the level of solar illumination. Since both time series include a repeating seasonal variation, their cross-correlation will show a cyclical correlation at a relatively low level. Panel (b) of fig. 11 shows the cross correlation for 40 days around the zero lag; there is a clear positive correlation at the zero lag for 90 km and a smaller negative correlation for 70 km. Variances at 80 km show little evidence of a relationship with geomagnetic activity.

These observations can be explained as follows: During periods of high geomagnetic activity, there is an influx of high-energy particles into the mesosphere (e.g. Brasseur and Solomon, 2005). This means that at lower altitudes, where there is normally very little ionisation, the ionisation levels increase, and partial reflection of radio waves is stronger. As we have already seen, measured wind variance is related to the scatter quality, so an increased scatter quality corresponds to a lower measured variance at 70 km.

Increased ionisation levels at the lower altitudes also have the effect of absorbing radio waves that pass through, meaning that the quality of signal for radio waves partially reflected at higher altitudes is diminished. Thus, we see the inverse effect for data from 90 km: periods with increased geomagnetic activity correspond to an in- crease in measured variance at higher altitudes, as the amount of data decreases. The correlations seen at 70 and 90 km decay with lag times of about 5-10 days, suggesting that this is the time scale over which the ionisation levels return to normal after a geomagnetic event. This would be in line with studies of energetic precipitation driven by solar wind transients such as high speed solar wind streams (e.g. Kavanagh et al., 2012). This reflects the pattern of SNR seen in Kavanagh et al. (2018) at Rothera in response to increased precipitation where there is a reduction in data at high altitudes due to signal loss and a gain in data at the lower altitudes. This hints at an underlying relationship between variance and data quality (in terms of the amount of data seen).

## 5.  Discussion

In general the variance of the wind speed measured by MF radar has been taken to be an indicator of gravity wave activity (the wave structures themselves being too small to be resolved by the radar). Previous authors have used averaged variances to produce climatologies of gravity waves in the mesosphere and lower thermosphere; Hibbins et al. (2007) found an annual climatology of gravity wave activity at Rothera very similar to that displayed in fig. 9 and noted that this annual trend does not agree with the expected trend of increased activity during the equinoxes; this suggested that some other factor was in play.

Analysis of the distribution of the wind velocity from both Halley and Rothera show that they exhibit very similar behaviour (even with the differences in the radar frequencies and power levels), with the tail of the distribution following a Lorentz (or Cauchy) distribution. We cannot fully exclude the possibility that gravity wave activity is contributing to the correlations during periods of high axial ratio, but one would need to explain why the wave action would result in the observed outlier

distribution and an increased axial ratio. It is not clear to us that observed distributions of wave activity would produce this
result (e.g. Matsuda, et al., 2017).

Many studies use an arbitrary velocity limit to remove data that are deemed 'unphysical', but this presupposes that the processes that drive winds in the mesosphere are sufficiently well understood that we are confident in ignoring high speeds. This is fine if the only interest is relatively slowly changing phenomena such as tides and planetary waves; however the threshold chosen for the wind speed will influence the variance response. A better method might be to use the axial ratio

property itself to limit the data; as we have seen this is strongly linked to the velocity but is a fundamental property of the fitting mechanism and one could make a strong case for a limit that excludes likely unphysical correlations. This is an approach recommended by Brown (1992) who suggests an axial ratio limit of 5 along with a number of other limits related to the fitting process.

Fig. 6 showed the results of a monte-carlo simulation of the height distribution of variance for both radars, using a fit to the

tail of the observed data distribution to define the Lorentz parameters. The shape of the simulation with height matched the observed variance in the data well, providing evidence that the variance is dominated by Lorentzian noise. However the match was not perfect, which might suggest that gravity waves still play a role in the variance.

Fig. 7 showed that solar illumination plays a significant role in affecting the variance that also varies with altitude. The simple explanation for this is that the radar partially reflects from density structures in the ionized portion of the atmosphere (the D-

region of the ionosphere), sunlight is the dominant source of ionization and so reduced sunlight results in reduced scatter from the radar. This leads to higher variance in darkness relative to the sunlit times. Differences appear at sunset and sunrise due to ion chemistry effects where stable negative ions may be formed reducing the electron density at sun-rise relative to sunset for a given solar elevation angle (e.g. Collis and Rietveld, 1991). This could also go some way to explaining the distribution of variances with solar zenith angle presented in fig. 8; the wider distribution for negative elevation angles could be partly caused

by mixing values from pre-dawn and post dusk.

Fig. 11 showed the relationship between a measure of geomagnetic activity (the AE index) and the wind variance at selected altitudes at Halley. In this context the AE index is used as a proxy for increased ionisation due to energetic charged particle precipitation; after solar illumination this is the next strongest source of ionisation at high latitudes. However, the increased ionisation due to precipitation can be significantly higher than the background level from the sun, consequently it has a very

different effect on the radar signal. At low altitudes it can provide additional scattering sources but it also leads to increased attenuation of the radar signal such that there is reduced signal (Kavanagh et al., 2018). This is shown in fig. 12 where there is a small but positive correlation with variance at the higher altitudes, which transitions to a small negative correlation at lower altitudes.

An interesting aspect of this study is that although both the Halley and Rothera radars have similar overall wind distributions

(fig. 3) there are differences in their altitude response (fig. 6) and in the relationship between wind speed, variance and axial ratio (fig. 9). Thus different radars with different power levels, operating frequencies and other settings and rejection criteria

could behave in quite different ways than presented here. However, given the standard step of outlier removal and lack of discussion of outlier features and error distributions in the literature, no reason has been found to suggest that the observed results are limited to these data sets.

## 6. Summary and Conclusions

By examining the algorithm by which wind velocities are derived from the radar signal, properties of the error distribution are described. This analysis suggests that in some cases varying data quality may have been erroneously interpreted as gravity wave activity.

This study has examined the error distribution of velocities derived from the Full Correlation Analysis technique applied to spaced-receiver MF radars. It has revealed a number of important considerations, with particular reference to the interpretation of the variance of the winds.

1. Wind data obtained by FCA is subject to Lorentzian-distributed errors, and, due to the form of the calculation, the size of this error distribution is related to the observed level of anisotropy of the diffraction pattern (i.e. FCA elliptical contours axial ratio).

2. The FCA axial ratio and the error distribution change with time of day, season, and altitude; these changes seem to have been interpreted as real wind features in several previous studies over the past 30 years.

3. The change in the error distribution with altitude can be explained by differing scatter quality, due to the well-known changes in ion density with altitude.

4. Annual and diurnal components of the changes in the error distribution within each altitude can be explained by changes in ion density due to the daily cycle of photoionisation in the D-region of the ionosphere. This gives the seasonal pattern that has been erroneously interpreted purely as the result of gravity wave activity.

5. There is evidence that the influx of electrons due to geomagnetic activity accounts for additional features observed in the wind speed variance, especially at Halley. The details of this relationship, as well as its magnitude, remain to be explored further. This relationship is further evidence of a strong dependence on the analysis technique rather than a physical change in the small scale wind field.

This new understanding of the wind data has important implications for mesospheric wind measurements using MF radar and FCA, the full extent of which remains to be seen. In particular, this study considered data obtained from only two radars: the similarities or differences between data from different radars and these results should first be investigated to determine to what extent the observed features are particular to the radars, or universal between data sets.

Other directions for further work include an analysis of the process by which axial ratio changes, including whether this is a random process due to poor signal, or a physical response in the atmosphere. This could involve, for example, direct

comparison to the signal-to-noise ratio of the radar. Ideally, raw radar signals would be analysed to see the pattern properties resulting in the lag times and FCA parameters deduced.

Finally, the full extent of the impact of spurious wind speed measurements on analyses of MF radar data should be considered. If the level of error observed in this study turns out to be a common feature, there could potentially be impacts on other types of analyses, suggesting that careful quantification of the magnitude of this effect should be undertaken.

**Appendix**

Full Correlation Analysis (FCA) is a technique developed in the late 1980s at Adelaide University to obtain the bulk motion

of a generally anisotropic, time-evolving pattern probed by at least three sensors continuously in time. In the case of spaced-antenna radar observation of atmospheric winds, the pattern is that of radio wave reflections from the atmosphere, and the sensors are antennae recording the reflected radio wave signal.

In general, this method can use (with increasing levels of redundancy) an arbitrary number of sensors; here we show the simplest case, with just three antennae arranged in an L- shape. This is a reproduction of the algorithm as presented in Briggs

[6].

The setup is shown in Figure 13. Three sensors, S0, S1, and S2 are located along the orthogonal directions and each records the pattern strength at their local position continuously as a function of time. From these recorded signals f (x, y, t), correlation functions with lag time $\tau$, and across distances $\xi$ and $\eta$ in the x- and y-directions are calculated via:

$$\rho(\xi, \eta, \tau) = \frac{\langle (f(x,y,t)f(x+\xi,y+\eta,t+\tau) \rangle}{\langle f^2(x,y,t) \rangle} \qquad \text{(A1)}$$

Based on the spatial and temporal evolution of the pattern, there is a family of surfaces in the ($\xi, \eta, \tau$) plane – assumed to be ellipsoidal – that defines surfaces of constant correlation (the origin has $\rho = 1$, and the correlation strength decays in each direction away from the origin).

We define the coordinates ($x^l, y^l, t^l$) as those of a moving frame, stationary with respect to the overall drift of the pattern. In this frame, the correlation function will then be given by:

$$\rho(\xi', \eta', \tau') = \rho(A\xi'^2 + B\eta'^2 + 2H\xi'\eta' + K\tau^2) \qquad \text{(A2)}$$

Where $A, B, H,$ and $K$ are constants defining the shape of the pattern. The correlation function is constant at surfaces defining a tilted ellipse in the spatial dimensions (representing a generally anisotropic pattern), along with a term representing a decay in the time dimension.

If the pattern has bulk motion at speed V in the φ direction, a stationary observer's coordinates are defined relative to the moving observer by

$$x = x' + Vt \sin \phi = x' + V_x t \tag{A3}$$
$$y = y' + Vt \cos \phi = y' + V_y t \tag{A4}$$

Therefore, we substitute for $\xi^l$ and $\eta^l$ in Equation A2 to give


$$\rho(\xi, \eta, \tau) = \rho(A[\xi - V_x\tau]^2 + B[\eta - V_y\tau]^2 + 2H[\xi - V_x\tau][\eta - V_y\tau] + K\tau^2)$$

Rearranging and combining terms, this becomes

$\quad \rho(\xi, \eta, \tau) = \rho(A\xi^2 + B\eta^2 + 2H\xi\eta + C\tau^2 + 2F\xi\tau + 2G\eta\tau + K\tau^2) \tag{A5}$

Where we have defined F and G such that

$$F = -AV_x - HV_y \tag{A6}$$
$\quad G = -HV_x - BV_y \tag{A7}$

These are equations (2) and (3) in the main body of the paper.
Now, in order to determine the velocity components $V_x$ and $V_y$, we need to determine the coefficients of the ellipse $A, B, F, G,$ and $H$, to within a multiplicative constant. This can be done by considering the following five time
$\quad$ shifts obtained by cross- and auto-correlating the three signals obtained at $S_0$, $S_1$, and $S_2$:

1. $\tau_x$: the time shift at which the auto-correlation matches the cross-correlation between signals $S_0$ and $S_1$ at zero lag. From Equation A5, the auto- correlation is given by

$$\rho(\xi = 0, \eta = 0, \tau) = \rho(C\tau^2)$$

$\quad$ The cross-correlation is given by

$$\rho(\xi = \xi_0, \eta = 0, \tau = 0) = \rho(A\xi_0^2)$$

For these to equate the arguments of the correlation functions must be equal, so we have

$\quad \dfrac{A}{C} = \dfrac{\tau^2}{\xi_0^2} \tag{A8}$

2. $\tau_y$: the time shift at which the auto-correlation matches the cross-correlation between signals $S_0$ and $S_2$ at zero lag.
$\quad$ Similarly,

$$\frac{B}{C} = \frac{\tau_y^2}{\eta_0^2} \qquad\qquad\qquad\qquad (A9)$$

3. $\tau_{xy}$: the time shift at which the auto-correlation matches the cross-correlation between signals $S_1$ and $S_2$. Here, equating the arguments gives

$$\frac{H}{C} = \frac{\tau_{xy}^2}{2\xi_0\eta_0} = \frac{A\xi_0}{2C\eta_o} - \frac{B\eta_0}{2C\xi_0} \qquad\qquad (A10)$$

Since *A/C* and *B/C* have already been found, this is sufficient to obtain *H/C*

4. $\tau'_x$: the time shift at which the correlation between signals $S_0$ and $S_1$ is maximized. Equation A5 becomes

$$\rho(\xi = \xi_0, \eta = 0, \tau) = \rho(A\xi_0^2 + 2F\xi_0\tau + C\tau^2)$$

For the maximal time shift $\tau'_x$ we must have

$$\frac{\partial\rho}{\partial\tau} = (2F\xi_0 + 2C\tau'_x)\rho' = 0$$
And so we obtain

$$\frac{F}{C} = -\frac{\tau'_x}{\xi_0} \qquad\qquad\qquad\qquad (A11)$$

5. $\tau'_y$: the time shift at which the correlation between signals $S_0$ and $S_1$ is maximized. Similarly,

$$\frac{G}{C} = -\frac{\tau'_y}{\eta_0} \qquad\qquad\qquad\qquad (A12)$$

By substituting equations A8, A9, A10, A11 and A12 into equations A6 and A7 and solving the system of equations, the x- and y- components of the bulk drift velocity, $V_x$ and $V_y$ are then obtained.

**Data Availability**

Data from both MF radars are freely available from the UK Polar Data Centre (https://www.bas.ac.uk/data/uk-pdc/).

The AE index is available from the World Data Center for geomagnetism at Kyoto (http://wdc.kugi.kyoto-u.ac.jp/wdc/Sec3.html.).

## Author Contribution

MG carried out the bulk of the research in this paper as part of her final undergraduate research project, under the supervision of AJK. AJK is the PI for both radars used in this study. AJK prepared the manuscript from a written report by MG, who also contributed. Both MG and AJK developed the code to produce the figures, using MATLAB.

## Competing interests

The authors declare that they have no conflict of interest

## Acknowledgements

The Rothera MF radar is a joint project between the British Antarctic Survey and GATS Inc., Boulder, USA. Work on the MF radar at BAS is supported via NERC (NE/R016038/1). The Halley MF radar is wholly owned and operated by BAS. We would like to thank all of the staff who have wintered at Rothera and Halley station and in particular the electronics engineers who have been responsible for maintaining the MF radars. We would like to thank Neil Cobbett at BAS for ensuring the continuing operation of the MF radars and Genesis Software for their support. All data analysis was performed using code written in MATLAB (a property of Mathworks). The map in figure 1 was produced by the Mapping and Geographic Information Centre (MAGIC) at BAS.

## Refeences

Beldon C. L., and Mitchell, N. J.: Gravity wave-tidal inter- actions in the mesosphere and lower thermosphere over Rothera, Antarctica (68 S, 68 W). J. Geophys. Res., 115(18):D18101, 2010.

Brasseur G. P., and Solomon, S.: Aeronomy of the Middle Atmosphere. Chemistry and Physics of the Stratosphere and Mesosphere. Springer, 3rd edition, 2005.

Briggs. B. H.: The analysis of spaced sensor records by correlation techniques. In International Council of Scientific Unions Middle Atmosphere Program, Vol. 13, pp 166-186 (SEE N85-17452 08-46), 1984.

Brown, W. O. J.: MF radar interferometry, Ph.D. thesis, 312 pp., Univ. of Canterbury, Christchurch, N. Z., 1992

Collis, P. N., & Rietveld, M. T.: Mesospheric observations with the EISCAT UHF radar during polar cap absorption events: 1. Electron densities and negative ions. Ann. Geophys., 8, 809–824, 1990.

Dowdy A.J., Vincent, R. A., Igarashi, K. Murayama, Y., and Murphy, D. J.: A comparison of mean winds and gravity wave activity in the northern and southern polar MLT. Geophys. Res. Lett., 28(8):1475–1478, 2001.

Dowdy, A. J. Vincent, R. A., Tsutsumi, M., Igarashi, K. Murayama,Y. W. Singer,W. and Murphy D. J.: Polar mesosphere and lower thermosphere dynamics: 1. Mean wind and gravity wave climatologies. J. Geophys. Res. 112(D17):11685, 2001.

Espy, P. J. Hibbins, R. E., Swenson, G. E., Tang, J., Taylor, M. J., Riggin, D. M., and Fritts, D. C.: Regional variations of mesospheric gravity-wave momentum flux over Antarctica. Ann. Geophys., 24(1):81–88., 2006

Forbes, J. M.: Atmospheric tides between 80 km and 120 km. Advances in Space Research, 10(12), 127-140. COSPAR, 1990

Fritts, D. C.: Gravity wave saturation in the middle atmosphere: A review of theory and observations. Reviews of Geophysics and Space Physics, 22:275-308, 1984.

Gardner, F. F. and Pawsey, J. I. Study of ionospheric D-region using partial reflection, J. Atmos. Terr. Phys., 3, 321-344, 1953.

Hibbins, R. H., Espy, P. J. and Jarvis. M. J.: Quasi-biennial modulation of the semidiurnal tide in the upper mesosphere above Halley, Antarctica. Geophys. Res. Lett., 34(21), L21804, 2007.

Hocking, W. K.: A review of Mesosphere-Stratosphere- Troposphere (MST) radar developments and studies, circa 1997-2008. J. Atmos. Sol-Terr. Phys., 73(9):848–882, 2011.

Hocking, W. K., May, P. and Röttger. J.: Interpretation, Reliability and Accuracies of Parameters Deduced by the Spaced Antenna Method in Middle Atmosphere Applications. Pure and Applied Geophysics, 130(2-3):571–604, 1989.

Hoffmann, P., Rapp, M., Singer, W. and Keuer. D.: Trends of mesospheric gravity waves at northern middle latitudes during

summer. J. Geophys. Res. 116(D4):D10115, 2011.

Hoffmann, P., Becker, E., Singer, W., and Placke, M.: Seasonal variation of mesospheric waves at northern middle and high latitudes. J. Atmos. and Sol.-Terr. Phys., 72(14-15):1068– 1079, 2010.

Holdsworth D. A., and Reid, I. M.: The Buckland Park MF radar: routine observation scheme and velocity comparisons. Ann. Geophys., 22(11), 3815–3828, 2004.

Holzworth, R. H., and Meng, C.-I., Mathematical representation of the auroral oval, Geophys. Res. Lett., 2(9), 377-380, 1975

Isler, J. R. and Fritts D. C.: Gravity Wave Variability and Interaction with Lower-Frequency Motions in the Mesosphere and Lower Thermosphere over Hawaii. http://dx.doi.org/10.1175/1520- 0469(1996) 53(1):37–48, 1996.

Kavanagh, A. J., Cobbett, N., & Kirsch, P.: Radiation Belt slot region filling events: Sustained energetic precipitation into the mesosphere. J. Geophys. Res.: Space Physics, 123, 7999– 8020. https://doi.org/10.1029/2018JA025890, 2018.

Kavanagh, A. J., Honary, F., Donovan, E. F., Ulich, T., and Denton, M. H.: Key features of >30 keV electron precipitation during high speed solar wind streams: A superposed epoch analysis, J. Geophys. Res., 117, A00L09, doi:10.1029/2011JA017320, 2012.

Kelley, M. C.: The Earth's Ionosphere. Plasma Physics and Electrodynamics. Elsevier. 2009.

Manson, A. H., Meek, C. E., Teitelbaum, H., Vial, F., Schminder, R., Kiirschner, D., Smith, M. J., Fraser, G. J., and Clark, R.

R.: Climatologies of semi-diurnal and diurnal tides in the middle atmosphere (70-110 km) at middle latitudes ( 40- 55°). Journal of Atmospheric and Terrestrial Physics, 51, 579-593. 1989.

Matsuda, T. S., Nakamura, T., Ejiri, M. K., Tsutsumi, M., Tomikawa, Y., Taylor, M. J., Zhao, Y., Pautet, P.-D., Murphy, D. J., and Moffat-Griffin, T.: Characteristics of mesospheric gravity waves over Antarctica observed by Antarctic Gravity Wave

Instrument Network imagers using 3-D spectral analyses, J. Geophys. Res. Atmos., 122, 8969–8981, doi:10.1002/2016JD026217, 2017.

Meek, C. E., Reid, I. M., and Manson A. H.: Observations of mesospheric wind velocities: 2. Cross sections of power spectral density for 48-8 hours, 8-1 hours, and 1 hour to 10 min over 60-110 km for 1981. Rad. Sci., 20(6):1383–1402, 1985.

Mudelsee. M.: Climate time series analysis: classical statistical and bootstrap methods: Atmospheric and Oceanographic Sciences Library, v. 42, 474, 2010.

Nakamura, T., Tsuda T., Fukao, S., Kato, S., Manson, A. H., and Meek, C. E.: Comparative Observations of Short- Period Gravity-Waves (10-100-Min) in the Mesosphere in 1989 by Saskatoon Mf Radar (52-Degrees-N), Canada and the Mu Radar (35-Degrees-N), Japan. Rad. Sci., 28(5):729–746, 1993.

Press, W. H., Saul A Teukolsky, S. A., Vetterling, W. T., and Flannery, B. P. Numerical Recipes. Cambridge University Press, 3rd edition.

Price, G. D., Jacka, F., Vincent, R. A., and Burns, G. B.: The influence of geomagnetic activity on the upper mesosphere lower thermosphere in the auroral zone. II. Horizontal winds. J. Atmos. and Terr. Phys., 53(10), 923–947, 1991.

Reid, G. C.: The influence of electric fields on radar measurements of winds in the upper mesosphere. Radio Science, 18, 1028-1034, 1983.

Russell, C. T., and McPherron, R. L. (1973), Semiannual variation of geomagnetic activity, J. Geophys. Res., 78(1), 92– 108, doi:10.1029/JA078i001p00092.

Thayaparan, T., Hocking, W. K., and MacDougall J.: Observational evidence of tidal/gravity wave interactions using the UWO 2 MHz radar. Geophys. Res. Lett., 22 (4), 373–376, 1995.

Vanzandt, T. E., Fritts, D. C. and Vanzandt T. E.: Spectral Estimates of Gravity Wave Energy and Momentum Fluxes. Part I: Energy Dissipation, Acceleration, and Constraints. http://dx.doi.org/10.1175/1520- 0469(1993), 50(22), 3685–3694, 1993.

Vincent R. A. and Fritts D. C.: A Climatology of Gravity Wave Motions in the Mesopause Region at Adelaide, Australia. Journal of the Atmospheric Sciences, 44(4), 748–760, 1987.

Yamazaki, R., Igarashi, K., Nagayama, M., and Nishimuta, I.: The effect of signal saturation on wind velocity estimates using MF radar signals. Adv. Space Res., 25 (1):223–226, 2000.

Zhang, Y., Li, J., and Zhou, L.: The Relationship between Polar Vortex and Ozone Depletion in the Antarctic Stratosphere during the Period 1979–2016, Adv. in Meteorology, 2017, doi: 10.1155/2017/3078079, 2017.

|  | **Rothera** | **Halley** |
|---|---|---|
| Radar power / kW | 25 | 100 |
| Frequency / MHz | 1.98 | 2.7 |
| Average time step / s | 100 | 50 |
| Altitude range / km | 56.5-102.5 | 52-98 |

| Years available | 2002-current | 2012-2016 |
| --- | --- | --- |

**Table 1. Basic information on the operation characteristics of the two radars.**

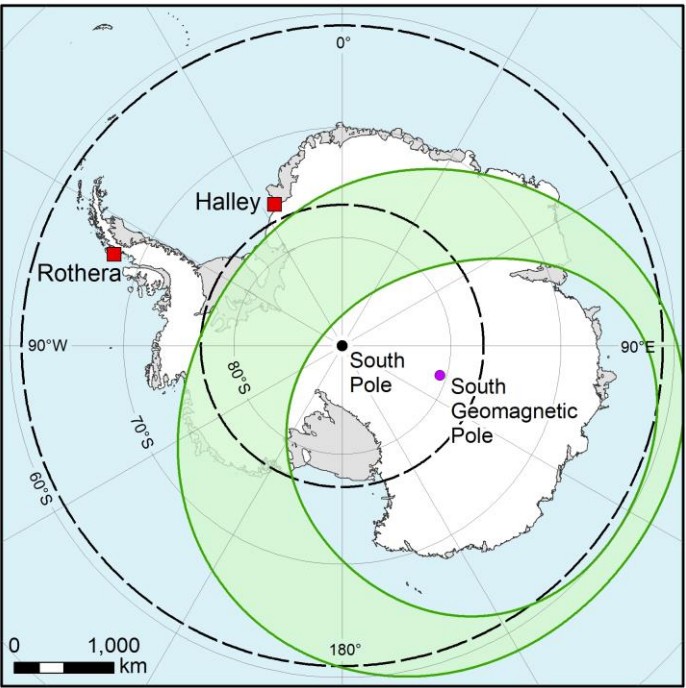

**Figure 1: Location of the two MF radars used in this study (Halley and Rothera) marked as red squares, with the locations of the geocentric (black) and geomagnetic (purple) south poles. The dashed black lines give estimates of the extent of the polar vortex from May (inner) to August (outer) (from Zhang et al., 2017). The green shaded region shows the statistical location of the auroral oval for quiet (Kp=3) geomagnetic activity (Holzworth and Meng, 1975)**


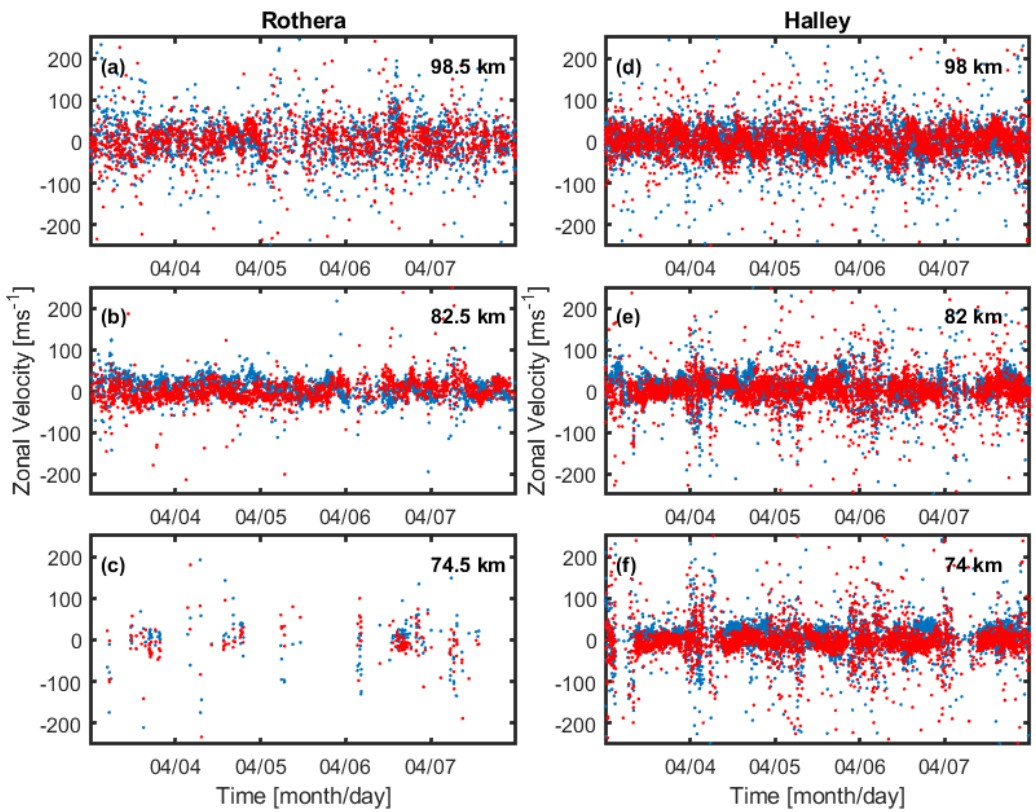

**Figure 2. Sample data from the Rothera (left) and Halley (right) radars for three comparable heights from 3-7 April 2013. Blue dots represent zonal winds, red dots represent meridional winds. Large variability can be seen in each plot. At the lowest altitude, Rothera (c) is experiencing a loss of scatter due to a weaker (than Halley (f)) returned signal.**

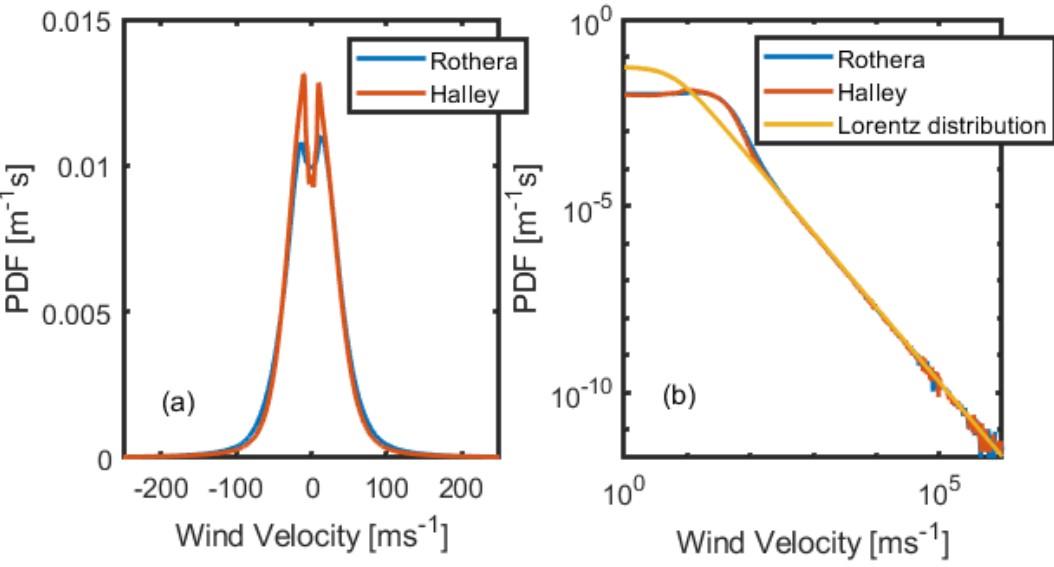


**Figure 3. Panel (a): probability distribution function of the combined (zonal and meridional) wind velocities at Halley (red) and Rothera (blue), for the entire data set. The double hump is due to the tidal influence over the winds. Panel (b): The same distributions plotted on a log-log scale to illustrate the long tails of the distributions, with a Lorentz distribution (b = 5.7) fitted to the data beyond ~300 ms⁻¹ (yellow).**

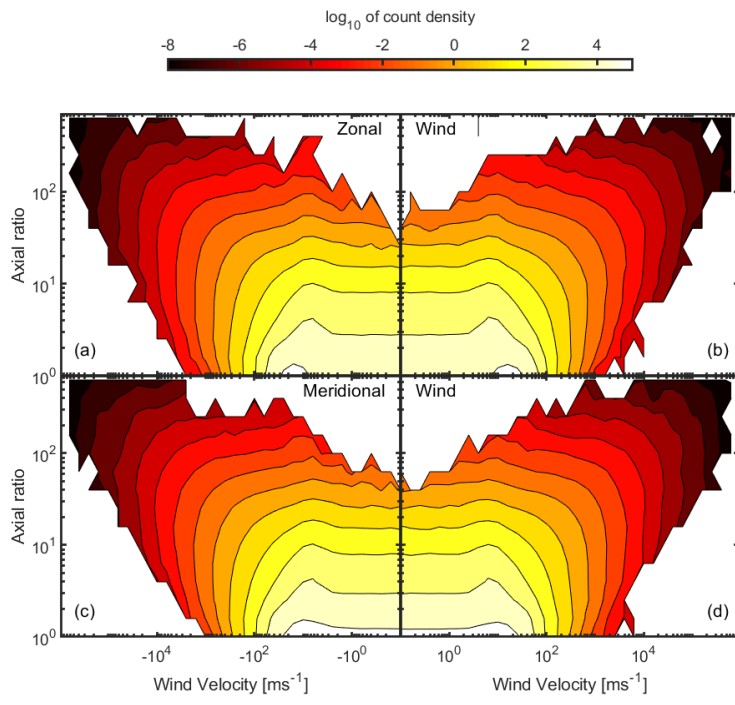


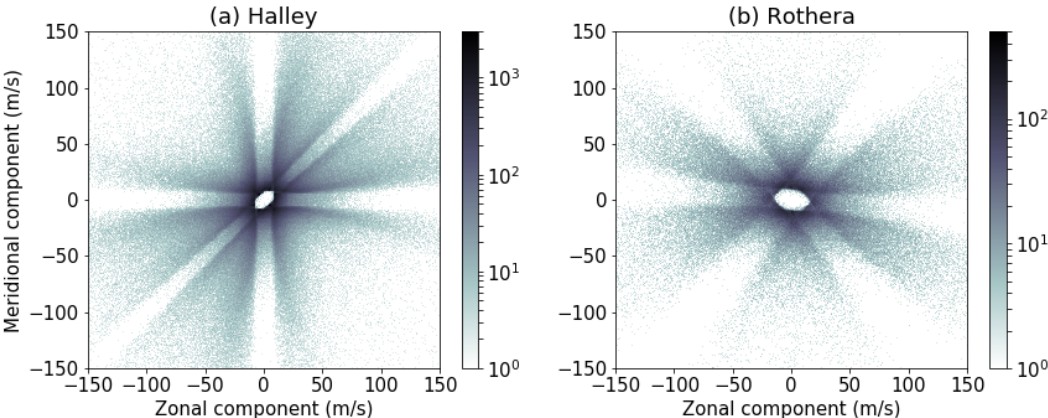

**Figure 4. Wind speed versus axial ratio for 28 million data points, taken from the Rothera MF radar from 2002-2016. The four panels represent the axial ratio binned by the 4 direction (a) west, (b) east, (c) south and (d) north. Note that the colour scale is the logarithm of the count density.**

**Figure 5. Two-dimensional density plots of wind velocity measurements with ellipse axial ratio R>5. Data points from all altitudes are included. The data from both radars show qualitatively similar patterns which seem to be artefacts of the 3-antenna arrangement and full-correlation analysis measurement technique.**

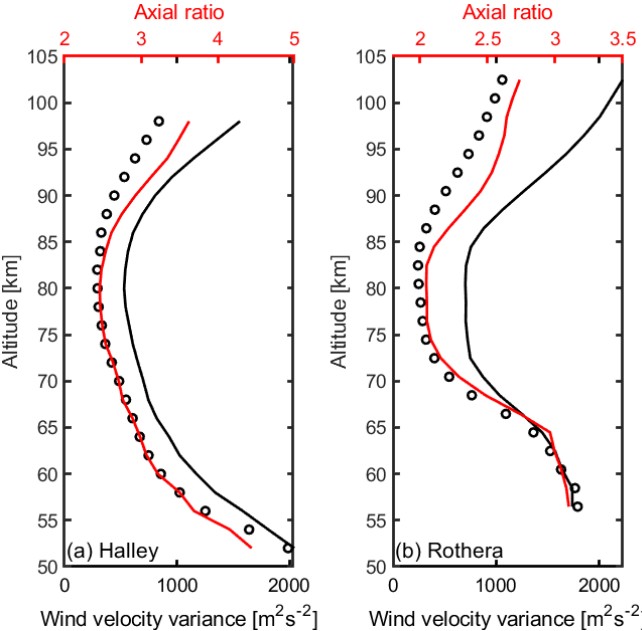

**Figure 6. Vertical profiles of observed hourly variances in zonal wind data, excluding velocities over 150 ms[-1] (black), the expected hourly variance based on the observed number of outliers (black circles), the mean axial ratio (red).**

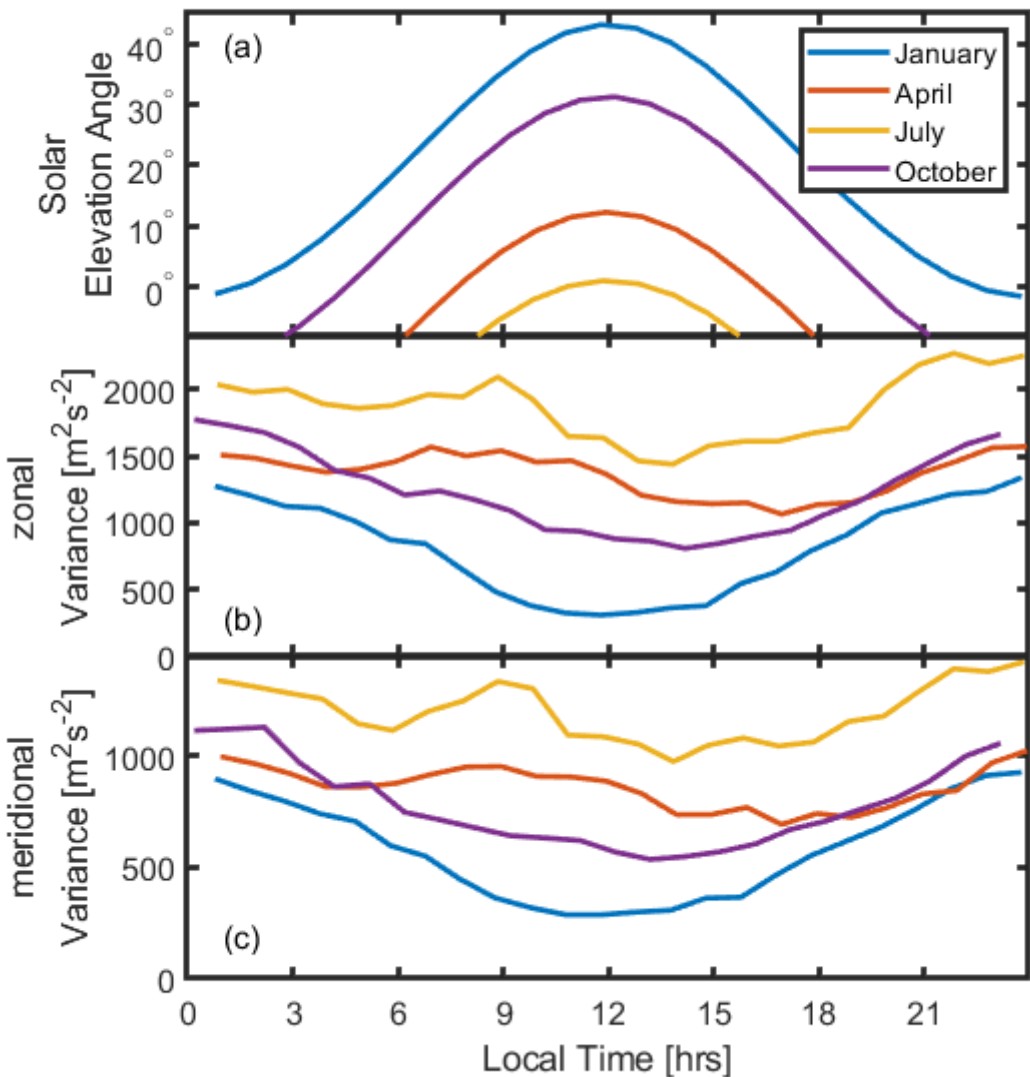

**Figure7: (a) The average solar elevation angle for the 4 months indicated as a function of local time at Rothera, (b) the average daily cycle of zonal variance averaged over range gates from 88.5 to 90.5 km km for the same months: January (blue), April (red), July (yellow) and October (purple), (c) the same but for meridional variance.**


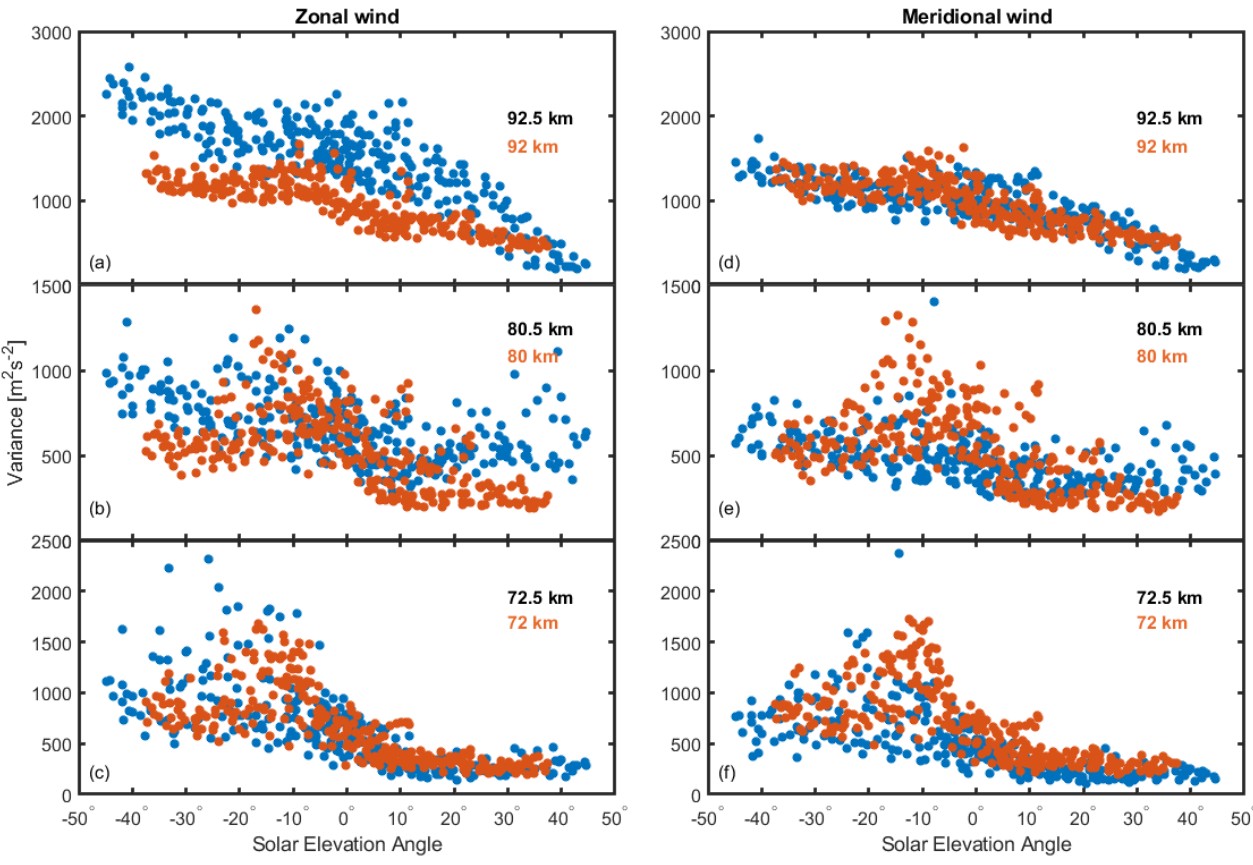

**Figure 8. Zonal (left) and meridional (right) wind variance from the Rothera (blue) and Halley (Orange) radar for three altitudes, as a function of solar elevation angle.**

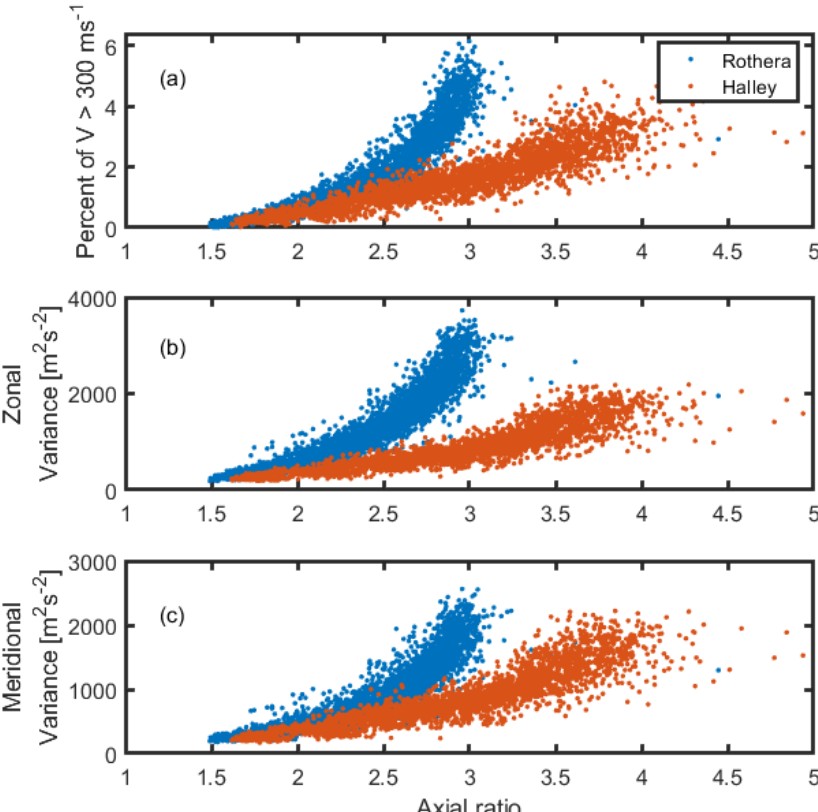

**Figure 9. The relationship between axial ratio and (a) outliers and (b) hourly variance of the zonal wind and (c) hourly variance of the meridional wind. Each point represents a different hour, month and altitude, to isolate the effect of differing error levels. Only altitudes over 80 km are shown. Meridional and zonal winds show substantially the same relationship, with slightly smaller variances in the meridional direction (reflective of the lower wind values).**

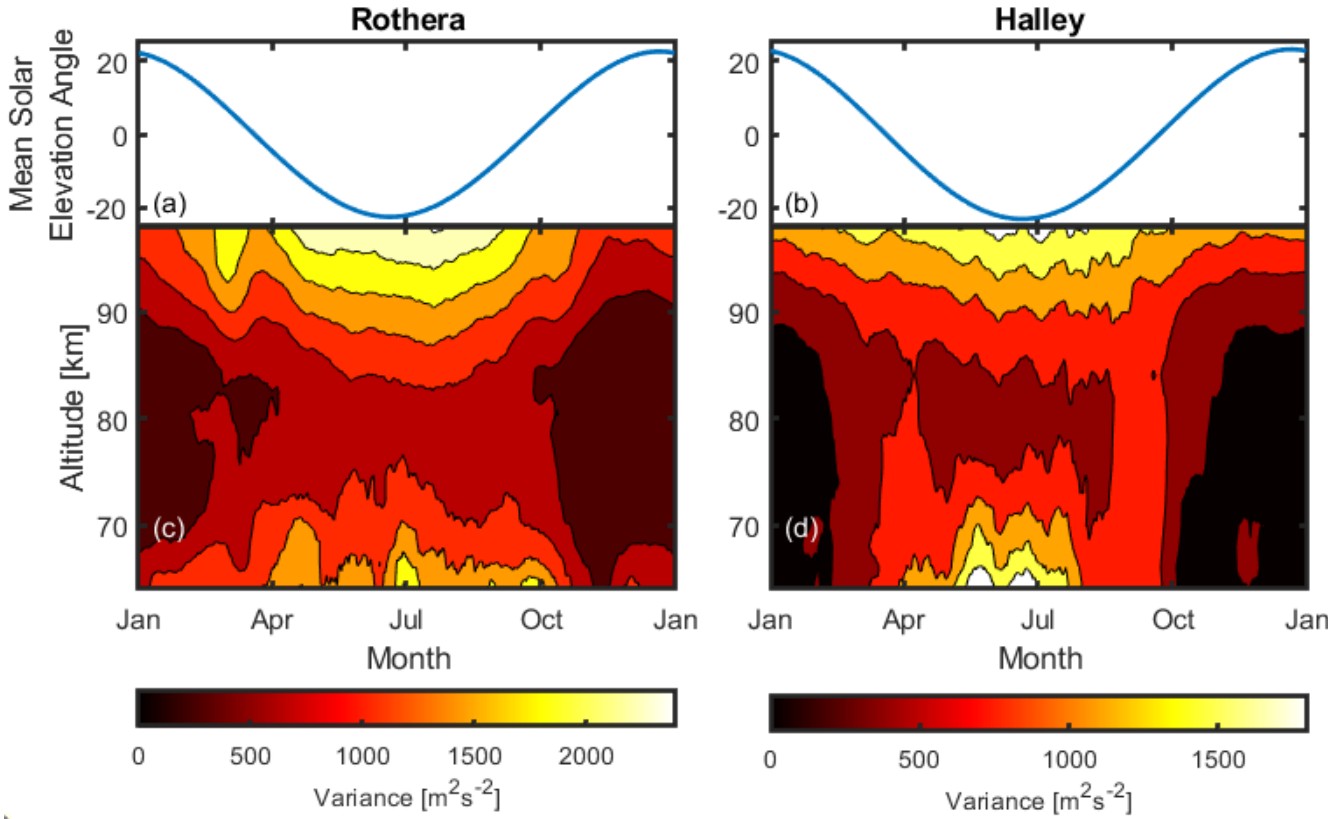

**Figure 10. Trends in variance over the course of the year at Rothera (c) and Halley (d) with the daily mean angle of the sun above the horizon ((a) and (b)). Mean variances were calculated for each day and the result smoothed using a 15-day running mean. Note the difference in colour scale between the two plots.**


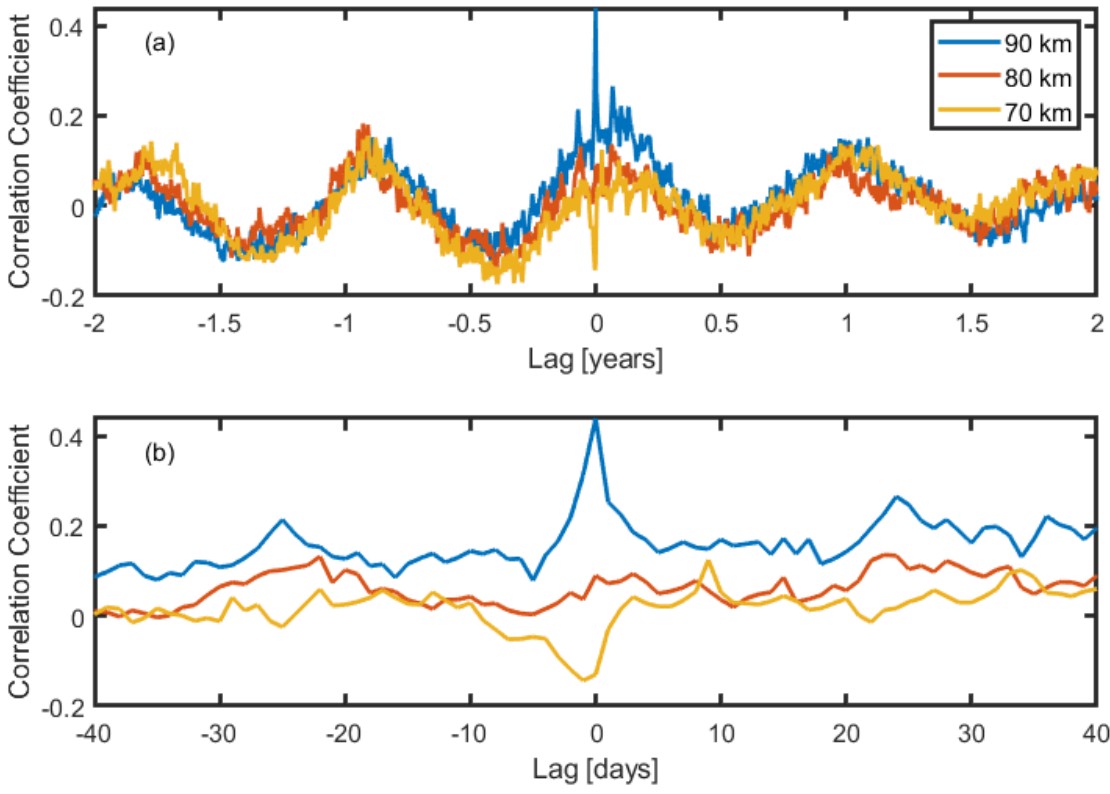

**Figure 11.** Correlation coefficients between daily averaged AE index and zonal wind variance observed at Halley shown at a range of lag times. Panel (a) shows the long term correlation: the sinusoidal nature of the correlation shows a seasonal cycle. Peaks are seen at zero lag at 70 and 90 km, suggesting a relationship beyond the seasonal variations. In panel (b) the central peaks are shown. A distinct correlation at zero lag is seen for some altitudes, positive at 90 km and negative at 70 km.

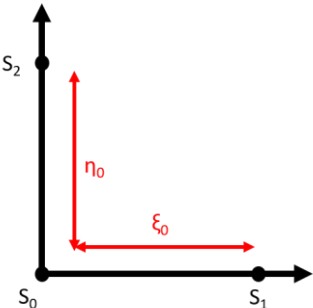

**Figure 11.** A schematic of the FCA setup. The pattern being probed is assumed to decay in the x- and y-direction in a generally anisotropic way, as well as evolving in time and travelling with a bulk motion. Three sensors S0, S1, and S2 are located at the origin,

**a distance $\xi_0$ away in the x-direction, and a distance $\eta_0$ away in the y-direction respectively (these represent the antennae for radar applications). Each sensor records the local pattern strength continuously in time.**