# Peer review of "Mesospheric winds measured by MF radar with Full Correlation Analysis: error properties and impacts on studies of wind variance"

_Geoscientific Instrumentation, Methods and Data Systems, 2019_

## Referee Comment (RC1) · Anonymous Referee #1 · 31 Dec 2019

The paper entitled "Mesospheric winds measured by MF radar with Full Correlation Analysis: error properties and impacts on studies of wind variance" by Gibbins and Kavanagh has for purpose to determine what could be the sources of errors measured by MF radars in Antarctica. The main result is that the occurrence of velocity outliers and the width of the velocity distribution recorded by MF radar have been possibly wrongly attributed to the influence of gravity waves but could be more likely be attributed to the anisotropic received signal pattern of the radar. The paper also states that the magnitude of the error distribution is related to solar insolation and geomagnetic activity.

[Figure]

Although the paper is showing interesting initial results which seem physically meaningful, the paper is quite brief and some sections are not always fully described nor the figures. Finally, the data analysis presented seems incomplete. I wish to see a revision of the paper before being able to accept it for publication.

First of all, as the paper is mainly based on finding the source of errors associated with the Full Correlation Analysis applied to MF radar, I would have liked to see this method better explained. Although this method is explained in full detail in Briggs et al. (1984), this paper is quite difficult to find and it would rather be useful for a reader to have at least the basic principles of such a method reminded in the present paper. As the paper is not too long, I do not think it would be much of a problem to add a section about this method.

Second, one of the central figure of the paper is missing: that is Figure 9. As this figure is supposed to fully confirm that there is a strong correlation between axial ratio and both number of outliers and hourly variance and thus that the radar data quality are the cause of the large wind values, I would really like to see this figure before accepting this paper.

Third, I would like to see explained why the correlation analysis made with respect to solar insolation and geomagnetic activity are only performed with zonal winds. This is not explained anywhere and sometimes, it is not even clear if this is only zonal winds or combined zonal and meridional winds which are presented, especially in figure 10 where it is not stated neither in the captions nor in the associated text. If for example, meridional winds naturally show a smaller signal variance then it is worth stating it.

Finally, I would like to see better explanation for Figure 11. Indeed, as the AE index is a measure of the geomagnetic activity in the Northern hemisphere while the MF radars data analyzed in this paper are in the Southern Hemisphere, it will be good to better describe the seasonal correlation, in particular for which seasons correspond the maxima and the minima of the correlation coefficients and to explain physically these

maxima and minima.

Other question: On Figure 7, I am really surprised that the solar elevation angle with respect to local time is so different between April and October whereas these two months are very close to equinoxes. Are the authors sure of their calculation?

Few typos: Line 99: "for Rothera, the diurnal tide maximizes in Winter whilst the semi-diurnal peaks in winter": I wonder if there is not a problem in the seasons listed in this sentence, is it really always winter? Line 134: "The data in fig. 3..." should be replaced by "The data in fig. 4..."

———————————————

---

## Referee Comment (RC2) · Anonymous Referee #2 · 22 Jan 2020

Dear authors, Dear Editor,

The paper entitled "Mesospheric winds measured by MF radar with Full Correlation Analysis: error properties and impacts on studies of wind variance" by Gibbins and Kavanagh aims at study the impact of different parameters (solar illumination, geomagnetic activity) on the variance of the zonal thermospheric winds. The paper is really interesting in many aspects, and I see a clear link with the ionised atmospheric community. I suggest the authors to take into account for minor comments (below) which could be useful for a non-expert reader.

[Figure]

Sincerely yours

Figure 1 : Add Auroral zone and polar vortex extents locations approximation.

Line 96 : "An oscillating signal can be seen, most strongly in panel (b) then (c)" We can't see any oscillation on (c) picture due to missing data. I guess the authors wanted to mention (e). Additionally, to highlight the semi-diurnal oscillation mention, it could be interesting to add on the plot the estimated oscillation (sliding-median or least-square adjustment). It could be also interesting to have the meridional in the same plots (in red for example).

Fig 3 : is it based on the entire data set or only on the 5 days in April 2013. Can you explain the larger wind speed in Halley compare to Rothera.

Line 134-136 : "The data in fig. 3 are from all altitude ranges from the Rothera radar: the relationship between velocity and axial ratio appears to be independent of altitude for the range of heights that the radars measure." I don't understand this statement as fig 3 (nor fig 4) is dependent on the altitude. Could the authors explain a little bit more on this point ?

Line 146, 147, . . . : Replace H2 by 2.H or something similar instead of H2 which is confusing.

Figure 6 and associated discussion: What is the period of time taking into account to produce this plots (especially the black circles). Is it years, month or a typical day or hour ? I suppose that the variance at a certain altitude "varies" with time, seasons, . . .. If not, could you add a sentence on this for a non-expert.

L233 : "This is interpreted as a response to the changing levels of ionisation". In the ionosphere, we call it Weddell Sea Anomaly, which corresponds to a maximum of electron density at local midnight during summer season. This paper "Chang, L. C., Liu, H., Miyoshi, Y., Chen, C., Chang, F., Lin, C., Liu, J. and Sun, Y. ( 2015), Structure and origins of the Weddell Sea Anomaly from tidal and planetary wave signatures in

FORMOSAT‐3/COSMIC observations and GAIA GCM simulations. J. Geophys. Res. Space Physics, 120: 1325– 1340. doi: 10.1002/2014JA020752." This could be interesting to discuss in the paper to make the link between neutral and ionised atmosphere. In theory is that the maximum electron density at local midnight is due to thermospheric winds.

Fig 8 and corresponding discussion: The solar Zenith Angle (SZA) stands generally for the angle between the zenith of the location considered and the sun angle with respect to this zenith. So 0° means at the Zenith, and +/-90° at the horizon. The authors should adapt the figure and the caption with respect to this rule as they mention that "the solar zenith angle (90 − solar elevation angle)" while from figure 7 top, the SZA should be in between 90-40=50° and 90-(-5)=95°. I suggest the author to SZA for zenith = 0° and below the horizon + or - 90° and + corresponding to the sunrise and sunset respectively. Figure 10: Do you observed the same patterns above Halley? It could be interesting to have both in case they behave in different ways.

AE comparison with zonal wind variance: this part is interesting in terms of climatology (seasonal correlations) but also in term of altitude impact. However, the correlations ranged between 0 and +/-0.2 which is really small. The authors should explain how we can be sure about the conclusions with small correlation coefficients.

---

## Author Comment (AC1) · 26 Feb 2020

PLease see my response in the supplement

Please also note the supplement to this comment:
https://www.geosci-instrum-method-data-syst-discuss.net/gi-2019-34/gi-2019-34-AC1-supplement.pdf

---

## Author Response (AR1)

Response to the referees

We would like to thank both referees for taking the time to review our manuscript and express our appreciation. The comments and suggestions they provided have allowed us to address errors and oversights in the text and to improve the manuscript.

We provide our responses to the reviewers below with their comments in italics and our response in bold. Changes to the manuscript are shown in plain text. Following this is the revised version of the manuscript with track
* * *
Referee 1:
* * *
Referee: *"First of all, as the paper is mainly based on finding the source of errors associated with the Full Correlation Analysis applied to MF radar, I would have liked to see this method better explained. Although this method is explained in full detail in Briggs et al. (1984), this paper is quite difficult to find and it would rather be useful for a reader to have at least the basic principles of such a method reminded in the present paper. As the paper is not too long, I do not think it would be much of a problem to add a section about this method."*

**We have inserted a basic description of the Full Correlation Analysis as an appendix and referred to it within the main body of the paper. We felt that inserting a section into the main body of the paper would disrupt the flow. We hope this is acceptable.**

Referee: *"Second, one of the central figure of the paper is missing: that is Figure 9. As this figure is supposed to fully confirm that there is a strong correlation between axial ratio and both number of outliers and hourly variance and thus that the radar data quality are the cause of the large wind values, I would really like to see this figure before accepting this paper."*

**Our apologies over the missing figure 9, when we first uploaded the manuscript it was missing, the editorial staff quickly informed us and we uploaded a new copy (which is downloadable from the site as the discussion paper); however clearly you were sent the original. A new version of the figure including both zonal and meridional winds has been included in the manuscript and is shown below.**

[Figure]

Figure 9. The relationship between axial ratio and (a) outliers and (b) hourly variance of the zonal wind and (c) hourly variance of the meridional wind. Each point represents a different hour, month and altitude, to isolate the effect of differing error levels. Only altitudes over 80 km are shown. Meridonal and zonal winds show substantially the same relationship, with slightly smaller variances in the meridional direction (reflective of the lower wind values).

Referee: *"Third, I would like to see explained why the correlation analysis made with respect to solar insolation and geomagnetic activity are only performed with zonal winds. This is not explained anywhere and sometimes, it is not even clear if this is only zonal winds or combined zonal and meridional wind, which are presented, especially in figure 10 where it is not stated neither in the captions nor in the associated text. If for example, meridional winds naturally show a smaller signal variance then it is worth stating it."*

**Thank you for pointing this out, this was a significant oversight on my part. We have included a more detailed description of the analysis in the manuscript and made sure that we indicate which data are presented in each case. In fact, we did the analysis with both the zonal and meridional winds and found similar results for both. We had opted to show just the zonal winds to limit the size of the plots; however, following the reviewer's comments we have included both wind directions in some of the plots and made sure we reference both in the text of the manuscript.**

**Figures 2, 7, 8, 9 and 10 now include separate zonal and meridional wind (as well as figures 4 and 5, which already did).**

Referee: *"Finally, I would like to see better explanation for Figure 11. Indeed, as the AE index is a measure of the geomagnetic activity in the Northern hemisphere while the MF radars data analyzed in this paper are in the Southern Hemisphere, it will be good to better describe the seasonal correlation, in particular for which seasons correspond the maxima and the minima of the correlation coefficients and to explain physically these and minima."*

**The referee is correct that the AE index is derived from northern magnetic stations. Furthermore it is a global scale (or perhaps half-global scale) measurement since it is averaged across those stations spread in longitude. Its purpose is to provide an indication of geomagnetic activity – driven primarily by the substorm cycle. Substorms are global phenomena and although there can be quite drastic differences in the local scale structure, magnitude and positioning of auroral forms (and the underlying magnetic topology), between the poles, in a statistical sense the AE index will still be representative of geomagnetic activity in the south. We have updated the text to improve the explanation, and we have modified old figures 11 and 12 into a single figure. We have also altered the analysis slightly to concentrate on the annual cycle and the effects around the zero lag,**

"To probe this relationship the Auroral Electroject (AE) index is used as a measure of geomagnetic activity. This index is derived from geomagnetic variations in the horizontal component of the magnetic field observed by 10 to 13 stations in the auroral zone in the northern hemisphere. The AE index is the difference between the largest and smallest values detected by these stations, produced at 1-minute resolution. It responds most strongly to the substorm cycle, where energy is loaded in the magnetotail from the solar wind, and then released earthward generating the auroral electroject and auroral displays. Although there can be quite drastic differences in the local scale structure, magnitude and positioning of auroral forms (and the underlying magnetic topology), between the poles, in a statistical sense the AE index will still be representative of geomagnetic activity in the south.

Figure 11, panel (a), shows the cross correlation between the daily averaged AE index and the daily averaged zonal wind variance measured at Halley at three altitudes: 90 km, 80 km and 70 km. Each of the data sets have been normalized such that their autocorrelations equal one at the zero lag and lie between 1and -1. At each altitude there is an annual cycle in the correlation, though the value of the coefficient is relatively small (<0.2). This cycle is due to the seasonal variations of both the variance and the AE index; the variability of the AE index is driven by changes in solar wind activity, but the coupling to Earth's magnetic environment has a seasonal component known as the Russell-McPherron effect (Russell and McPherron, 1973), whereby the coupling maximises around the equinoxes. Figure 10 illustrated that there is a seasonal pattern in the variance, which matches the level of solar illumination. Since both time series include a repeating seasonal variation, their cross-correlation will show a cyclical correlation at a relatively low level. Panel (b) of fig. 11 shows the cross correlation for 40 days around the zero lag; there is a clear positive correlation at the zero lag for 90 km and a smaller negative correlation for 70 km. Variances at 80 km show little evidence of a relationship with geomagnetic activity.

These observations can be explained as follows: During periods of high geomagnetic activity, there is an influx of high-energy particles into the mesosphere (e.g. Brasseur and Solomon, 2005). This means that at lower altitudes, where there is normally very little ionisation, the ionisation levels increase, and partial reflection of radio waves is stronger. As we have already seen, measured wind variance is related to the scatter quality, so an increased scatter quality corresponds to a lower measured variance at 60 km.

Increased ionisation levels at the lower altitudes also have the effect of absorbing radio waves that pass through, meaning that the quality of signal for radio waves partially reflected at higher altitudes is diminished. Thus, we see the inverse effect for data from 90 km: periods with increased geomagnetic activity correspond to an in- crease in measured variance at higher altitudes, as the amount of data decreases. The correlations seen at 70 and 90 km decay with lag times of about 5-10 days, suggesting that this is the time scale over which the ionisation levels return to normal after a geomagnetic event. This would be in line with studies of energetic precipitation driven by solar wind transients such as high speed solar wind streams (e.g. Kavanagh et al., 2012). This reflects the pattern of SNR seen in (Kavanagh et al 2018) at Rothera in response to increased precipitation where there is a reduction in data at high altitudes due to signal loss and a gain in data at the lower altitudes. This hints at an underlying relationship between variance and data quality (in terms of the amount of data seen)."

[Figure]

Figure 11. Correlation coefficients between daily averaged AE index and zonal wind variance observed at Halley shown at a range of lag times. Panel (a) shows the long term correlation: the sinusoidal nature of the correlation shows a seasonal cycle. Peaks are seen at zero lag at 70 and 90 km, suggesting a relationship beyond the seasonal variations. In panel (b) the central peaks are shown A distinct correlation at zero lag is seen for some altitudes, positive at 90 km and negative at 70 km.

**I would like to add that following the results of this paper, it is my intention to have a student project examining the response of the radar to local magnetic variations in more detail. But that is an additional significant amount of work designed to build upon this work and as such is beyond the scope of this paper.**

Referee: "Other question: On Figure 7, I am really surprised that the solar elevation angle with respect to local time is so different between April and October whereas these two months are very close to equinoxes. Are the authors sure of their calculation? "

**We have checked the calculations and are happy with them. To reassure the referee we have included some additional plots below. A factor to remember is that the plots represent averages across a whole month and so we are mixing data that is quite some time away from equinox in both cases**

**Figure R1.1 (a) shows the monthly average of the solar elevation angle as in Figure 7 of the manuscript, except that April and October have been replaced with the equinox months of March (red) and September (purple). Note that the solar elevation angles are much closer. It is worth noting the much more similar variance values for those two periods displayed in (b). Figure R1.2 is the same but instead of monthly averages we have averaged 28days surrounding the equinoxes; in this case the red and purple lines almost completely overlie one another, and again the corresponding variance curves are much more similar**

[Figure]

[Figure]

Fig. R1.1 As figure7 in the discussion paper but using March and September instead of April and October.

Fig R1.2 As figure R1.1 but with 28 day averages centred around the equinoxes.

Referee: *Line 99: "for Rothera, the diurnal tide maximizes in Winter whilst the semi-diurnal peaks in winter": I wonder if there is not a problem in the seasons listed in this sentence, is it really always winter?*

**Line 99: the reviewer is quite correct the first 'winter' should have been 'summer'**

*Line 134: "The data in fig. 3..." should be replaced by "The data in fig. 4..."*

**Line 134: corrected.**
* * *
Referee2:
* * *
Referee: *"Figure 1: Add Auroral zone and polar vortex extents locations approximation."*

**We have done this, by including dashed lines indicating the variable location of the edge of the vortex, and another shaded region showing an approximate auroral oval for quiet to moderate geomagnetic activity.**

[Figure]

Figure 1: Location of the two MF radars used in this study (Halley and Rothera) marked as red squares, with the locations of the geocentric (black) and geomagnetic (purple) south poles. The dashed black lines give estimates of the extent of the polar vortex from May (inner) to August (outer) (from Zhang et al., 2017). The green shaded region shows the statistical location of the auroral oval for quiet (Kp=3) geomagnetic activity (Holzworth and Meng, 1975)

"Figure 1 shows the locations of the two stations (red squares). The dashed black circles represent estimates of the statistical location of the edge of the polar vortex through winter from May (inner circle) to August (outer circle) (taken from Zhang et al., 2017). The shaded green region represents the extent of the quiet-time auroral oval (determined from Holzworth and Meng, 1975), where we have assumed local midnight lies between Rothera and Halley for the purposes of illustration."

Referee: "*Line 96 : "An oscillating signal can be seen, most strongly in panel (b) then (c)" We can't see any oscillation on (c) picture due to missing data. I guess the authors wanted to mention (e). Additionally, to highlight the semi-diurnal oscillation mention, it could be interesting to add on the*

*plot the estimated oscillation (sliding-median or least-square adjustment). It could be also interesting to have the meridional in the same plots (in red for example)."*

**The text should have read 'most strongly in panels (b) and (e); this has been corrected. We think that adding both a line to show the oscillation AND the meridional data makes the plot too confusing. Therefore, we have included just the meridional data since the identification of the tide is not actually important for the paper at this stage and the Halley data actually makes it stand out more clearly.**

"Figure 2 shows example time series of the wind data from (a) Halley and (b) Rothera for three altitude gates (98 km, 82 km and 74 km). Due to a timing issue at Rothera the range gates are offset by ~4.5 km. Blue (red) dots represent the zonal (meridional) winds. These illustrate some of the inherent properties of the data. Most measurements lie between -100 and 100 ms$^{-1}$ for this interval (87% of the data at Rothera, 76% at Halley) with seemingly random outliers. An oscillating signal is present in panels (b), (d) and (e); this is the semi-diurnal tide, which maximises in the high mesosphere. Tides are a major component of mesospheric winds in both the zonal and meridional directions. The dominance of tidal modes depends on location and time of year: for Halley the magnitudes of the diurnal and semidiurnal tides maximise in the summer months (Hibbins et al., 2006); for Rothera, the diurnal tide maximises in summer whilst the semi-diurnal peaks in winter (Hibbins et al., 2007). The tides tend to have wind amplitudes of a few tens of m/s, which can be additive depending on the phases of the tides. This still leaves a considerable amount of data that would be described as 'outliers'."

[Figure]

Figure 2. Sample data from the Rothera (left) and Halley (right) radars for three heights from 3-7 April 2013. Blue dots represent zonal winds, red dots represent meridional winds. Large variability can be seen in each plot. At the lowest altitude, Rothera (c) is experiencing a loss of scatter due to a weaker (than Halley (f)) returned signal.

Referee "*Fig 3 : is it based on the entire data set or only on the 5 days in April 2013. Can you explain the larger wind speed in Halley compare to Rothera.* "

**This is based on the entire data set (though Halley has a lot less data than Rothera), we have made this clear in the figure caption and the text. However, the wind speed is not appreciably larger in Halley than at Rothera according to the figure (speed is on the x-axis). Actually, the wind at Rothera is slightly faster than at Halley (particularly close to 50-100 m/s). We wonder if the referee means why the humps of the distributions are larger in Halley than in Rothera? This is the probability density function, such that one is more likely to find lower wind speeds at Halley than at Rothera, which might be due to the dominance of the tides. The numbers are very small such that the distributions are very similar.**

"Figure 3. Panel (a): probability distribution function of the combined (zonal and meridional) wind velocities at Halley (red) and Rothera (blue), for the entire data set. The double hump is due to the tidal influence over the winds. Panel (b): The same distributions plotted on a log-log scale to illustrate the long tails of the distributions, with a Lorentz distribution (b = 5.7) fitted to the data beyond ~300 ms$^{-1}$ (yellow)."

"Figure 3 shows the distribution of all wind speeds (zonal and meridional) measured by each of the two radars for the entire dataset. Panel (a) shows the probability distribution function (PDF) of a given velocity for data from Rothera (blue) and Halley (red). All velocity values are included: both zonal and meridional winds from all range gates. The PDF for each radar has a double hump, centred on the zero velocity, caused by the tidal nature of the wind; the rate of change of the wind will be lower where the tides have extrema, with a smaller number of data points appearing around zero velocity. The peaks represent some form of average magnitude of all significant tidal modes in the data set."

Referee: "*Line 134-136 : "The data in fig. 3 are from all altitude ranges from the Rothera radar: the relationship between velocity and axial ratio appears to be independent of altitude for the range of heights that the radars measure." I don't understand this statement as fig 3 (nor fig 4) is dependent on the altitude. Could the authors explain a little bit more on this point?*"

**Yes, this was a mistake, there should have been a line explaining that we had performed the analysis behind figure 4 (and figure 3) for limited altitude ranges and it had no significant effect on the result (apart from those height ranges where there was insufficient data).**

"The data in fig. 4 are from all altitude ranges from the Rothera radar. Limiting the altitude range produces the same results indicating that the relationship between velocity and axial ratio appears to be independent of altitude for the range of heights that the radars measure. The pattern remains the same when data from Halley are used."

Referee: "*Line 146, 147,...: Replace H2 by 2.H or something similar instead of H2 which is confusing.*"

**This was a mistake in the formatting – H2 should have been $H^2$. We have corrected this throughout the manuscript.**

Referee: *"Figure 6 and associated discussion: What is the period of time taking into account to produce this plots (especially the black circles). Is it years, month or a typical day or hour? I suppose that the variance at a certain altitude "varies" with time, seasons,.... If not, could you add a sentence on this for a non-expert."*

**The following lines are included in the text:**

"For each altitude simulated data were generated from equation 7 (where i = 1 million and is the length of the simulated time series) and the mean variance for velocities below 150 ms$^{-1}$ measured using a monte-carlo method with 100 iterations."

"The observed variance and axial ratio are averages of the hourly means that were calculated from data with wind speed < 150 m/s."

Referee: *"L233 : "This is interpreted as a response to the changing levels of ionisation". In the ionosphere, we call it Weddell Sea Anomaly, which corresponds to a maximum of electron density at local midnight during summer season. This paper "Chang, L. C.,Liu, H., Miyoshi, Y., Chen, C., Chang, F., Lin, C., Liu, J. and Sun, Y. ( 2015), Structure and origins of the Weddell Sea Anomaly from tidal and planetary wave signatures in FORMOSAT/COSMIC observations and GAIA GCM simulations. J. Geophys. Res. Space Physics, 120: 1325– 1340. doi: 10.1002/2014JA020752." This could be interesting to discuss in the paper to make the link between neutral and ionised atmosphere. In theory is that the maximum electron density at local midnight is due to thermospheric winds. "*

**We disagree with the referee. The Weddell Sea Anomaly, as indicated in the paper they provide is predominantly an F-region phenomenon, occurring at altitudes above 200 km. The radar is responding to changes in ionospheric density in the D region (below 100 km), where according to the figures in the recommended paper, the density perturbation from the tides is close to zero at the magnetic latitude of Rothera. The response of the variance in panel (b) of Figure 7 follows the change in solar elevation angle without needing to invoke the anomaly. The ionisation changes at those altitudes are predominantly dependent on solar illumination: the greater the amount of illumination (high elevation) the lower the variance and the vice versa.**

Referee*: "Fig 8 and corresponding discussion: The solar Zenith Angle (SZA) stands generally for the angle between the zenith of the location considered and the sun angle with respect to this zenith. So 0∘means at the Zenith, and +/-90∘at the horizon. The authors should adapt the figure and the caption with respect to this rule as they mention that "the solar zenith angle (90 – solar elevation angle)" while from figure 7 top, the SZA should be in between 90-40=50∘and 90-(-5) = 95∘. I suggest the author to SZA for zenith = 0∘and below the horizon + or - 90∘and + corresponding to the sunrise and sunset respectively".*

**This is a mistake that we made in the labelling that has carried over from an older version of the plot. We switched to using solar elevation angle to be consistent with the other figures and forgot to alter the caption and label. The correct x –axis label should be 'solar elevation angle' where 0**

**degrees represents the Sun's apparent position at the horizon, negative values indicate the sun is below the horizon and positive that it is above the horizon. This has now been corrected.**

Referee: *"Figure 10: Do you observed the same patterns above Halley? It could be interesting to have both in case they behave in different ways."*

**We do see a similar pattern at Halley. This is now shown as a separate panel in the figure. There are some differences, which we discuss. We also have cut the plots off at a slightly higher altitude due to the paucity of data at the lower range gates (which is seasonal) particularly at the lower power Rothera radar.**

[Figure]

Figure 10. Trends in variance over the course of the year at Rothera (c) and Halley (d) with the daily mean angle of the sun above the horizon ((a) and (b)). Mean variances were calculated for each day and the result smoothed using a 15-day running mean. Note the difference in colour scale between the two plots.

Referee: *"AE comparison with zonal wind variance: this part is interesting in terms of climatology (seasonal correlations) but also in term of altitude impact. However, the correlations ranged between 0 and +/-0.2 which is really small. The authors should explain how we can be sure about the conclusions with small correlation coefficients."*

**We have increased the discussion about the correlation between AE and the radar data and we have changed the analysis slightly to remove the distracting daily variations that is not important for the paper. This has partly had the effect of increasing the power of the cross correlation at the zero lag – this change is due to the daily average capturing more of the effect in a statistical sense than hourly values which will always show some additional variation. Please see our response to referee 1 on this topic.**

[revised manuscript text omitted]